# An Online Method for A Class of Distributionally Robust Optimization with Non-Convex Objectives

**Qi Qi**[†][*], **Zhishuai Guo**[†][*], **Yi Xu**[‡], **Rong Jin**[‡], **Tianbao Yang**[†]
[†] Department of Computer Science, University of Iowa, Iowa City, IA 52242
[‡] Machine Intelligence Technology, Alibaba Group
{qi-qi, zhishuai-guo, tianbao-yang}@uiowa.edu,{yixu, jinrong.jr}@alibaba-inc.com

## Abstract

In this paper, we propose a practical online method for solving a class of distributionally robust optimization (DRO) with non-convex objectives, which has important applications in machine learning for improving the robustness of neural networks. In the literature, most methods for solving DRO are based on stochastic primal-dual methods. However, primal-dual methods for DRO suffer from several drawbacks: (1) manipulating a high-dimensional dual variable corresponding to the size of data is time expensive; (2) they are not friendly to online learning where data is coming sequentially. To address these issues, we consider a class of DRO with an KL divergence regularization on the dual variables, transform the min-max problem into a compositional minimization problem, and propose practical duality-free online stochastic methods without requiring a large mini-batch size. We establish the state-of-the-art complexities of the proposed methods with and without a Polyak-Łojasiewicz (PL) condition of the objective. Empirical studies on large-scale deep learning tasks (i) demonstrate that our method can speed up the training by more than 2 times than baseline methods and save days of training time on a large-scale dataset with $\sim$ 265K images, and (ii) verify the supreme performance of DRO over Empirical Risk Minimization (ERM) on imbalanced datasets. Of independent interest, the proposed method can be also used for solving a family of stochastic compositional problems with state-of-the-art complexities.

## 1 Introduction

Distributionally robust optimization (DRO) has received tremendous attention in machine learning due to its capability to handle noisy data, adversarial data and imbalanced classification data [42, 33, 4]. Given a set of observed data $\{\mathbf{z}_1, \ldots, \mathbf{z}_n\}$, where $\mathbf{z}_i = (\mathbf{x}_i, y_i)$, a DRO formulation can be written as:

$$\min_{\mathbf{w} \in \mathbb{R}^d} \max_{\mathbf{p} \in \Delta_n} F_{\mathbf{p}}(\mathbf{w}) = \sum_{i=1}^{n} p_i \ell(\mathbf{w}; \mathbf{z}_i) - h(\mathbf{p}, \mathbf{1}/n) + r(\mathbf{w}), \tag{1}$$

where $\mathbf{w}$ denotes the model parameter, $\Delta_n = \{\mathbf{p} \in \mathbb{R}^n : \sum_i p_i = 1, p_i \geq 0\}$ denotes a $n$-dimensional simplex, $\ell(\mathbf{w}; \mathbf{z})$ denotes a loss function on data $\mathbf{z}$, $h(\mathbf{p}, \mathbf{1}/n)$ is a divergence measure between $\mathbf{p}$ and uniform probabilities $\mathbf{1}/n$, and $r(\mathbf{w})$ is convex regularizer of $\mathbf{w}$. When $\ell(\mathbf{w}; \mathbf{z})$ is a convex function (e.g., for learning a linear model), many stochastic primal-dual methods can be employed for solving the above min-max problem [35, 20, 49, 48, 32]. When $\ell(\mathbf{w}; \mathbf{z})$ is a non-convex function (e.g., for learning a deep neural network), some recent studies also proposed stochastic primal-dual methods [41, 48].

---

[*]The first two authors make equal contributions. Correspondence to tianbao-yang@uiowa.edu.

35th Conference on Neural Information Processing Systems (NeurIPS 2021), virtual.

However, stochastic primal-dual methods for solving DRO problems with a non-convex $\ell(\mathbf{w}; \mathbf{z})$ loss function (e.g., the predictive model is a deep neural network) suffer from several drawbacks. First, primal-dual methods need to maintain and update a high-dimensional dual variable $\mathbf{p} \in \mathbb{R}^n$ for large-scale data, whose memory cost is as high as $O(n)$ per-iteration. Second, existing primal-dual methods usually need to sample data according to probabilities $\mathbf{p}$ in order to update $\mathbf{w}$, which brings additional costs than random sampling. Although random sampling can be used for computing the stochastic gradient in terms of $\mathbf{w}$, the resulting stochastic gradient could have $n$-times larger variance than using non-uniform sampling according to $\mathbf{p}$ (please refer to the supplement for a simple illustration). Third, due to the constraint on $\mathbf{p} \in \Delta_n$, the min-max formulation (1) is not friendly to online learning in which the data is received sequentially and $n$ is rarely known in prior.

*Can we design an efficient online algorithm to address the DRO formulation (1) without dealing with $\mathbf{p} \in \mathbb{R}^n$ for a non-convex objective that is applicable to deep learning?*

To address this question, we restrict our attention to a family of DRO problems, in which the KL divergence $h(\mathbf{p}, \mathbf{1}/n) = \lambda \sum_i p_i \log(np_i)$ is used for regularizing the dual variables $\mathbf{p}$, where $\lambda > 0$ is a regularization parameter. We note that this consideration does not impose strong restriction to the modeling capability. It has been shown that for a family of divergence functions $h(\mathbf{p}, \mathbf{1}/n)$, different DRO formulations are statistically equivalent to a certain degree [12]. The proposed method is based on an equivalent minimization formulation for $h(\mathbf{p}, \mathbf{1}/n) = \lambda \sum_i p_i \log(np_i)$. In particular, by maximizing over $\mathbf{p}$ exactly, (1) is equivalent to

$$\min_{\mathbf{w} \in \mathbb{R}^d} \left\{ F_{dro}(\mathbf{w}) = \lambda \log \left( 1/n \sum_{i=1}^n \exp(\ell(\mathbf{w}; \mathbf{z}_i)/\lambda)) \right) + r(\mathbf{w}) \right\}. \tag{2}$$

In an online learning setting, we can consider a more general formulation:

$$\min_{\mathbf{w} \in \mathbb{R}^d} \left\{ F_{dro}(\mathbf{w}) = \lambda \log \left( \mathbb{E}_{\mathbf{z}} \exp \left( \ell(\mathbf{w}; \mathbf{z})/\lambda \right) \right) + r(\mathbf{w}) \right\}. \tag{3}$$

The above problem is an instance of stochastic compositional problems of the following form:

$$\min_{\mathbf{w} \in \mathbb{R}^d} F(\mathbf{w}) := f(\mathbb{E}_{\mathbf{z}}[g_{\mathbf{z}}(\mathbf{w})]) + r(\mathbf{w}), \tag{4}$$

by setting $f(s) = \lambda \log(s), s \geq 1$ and $g_{\mathbf{z}}(\mathbf{w}) = \exp(\ell(\mathbf{w}; \mathbf{z})/\lambda)$. Stochastic algorithms have been developed for solving the above compositional problems. [44] proposed the first stochastic algorithms for solving (4), which are easy to implement. However, their sample complexities are sub-optimal for solving (4). Recently, a series of works have tried to improve the convergence rate by using advanced variance reduction techniques (e.g., SVRG [19], SPIDER [13], SARAH [36]). However, most of them require using a mega mini-batch size in the order of $O(n)$ or $O(1/\epsilon)^2$ **at every iteration or many iterations** for updating $\mathbf{w}$, which hinders their applications on large-scale problems. In addition, these algorithms usually use a constant step size, which may harm the generalization performance.

This paper aims to develop more practical stochastic algorithms for solving (3) without suffering from the above issues in order to enable practitioners to explore the capability of DRO for deep learning with irregular data (e.g., imbalanced data, noisy data). To this end, we proposed an online stochastic method (COVER) and its restarted variant (RECOVER). We establish a state-of-the-art complexity of COVER for finding an $\epsilon$-stationary solution and a state-of-the-art complexity of RECOVER under a Polyak-Łojasiewicz (PL) condition of the problem. PL condition has been widely explored for developing practical optimization algorithms for deep learning [52]. Compared with other stochastic algorithms, the practical advantages of RECOVER are:

1. RECOVER is an online duality-free algorithm for addressing large-scale KL regularized DRO problem that is independent of the high dimensional dual variable $\mathbf{p} \in \mathbb{R}^n$, which makes it suitable for deep learning applications.

2. RECOVER also enjoys the benefits of stagewise training similar to existing stochastic methods for deep learning [52], i.e., the step size is decreased geometrically in a stagewise manner.

---

[2]$\epsilon$ is either the objective gap accuracy $F(\mathbf{w}) - \min F(\mathbf{w}) \leq \epsilon$ or the gradient norm square bound $\|\nabla F(\mathbf{w})\|^2 \leq \epsilon$

Table 1: Summary of properties of state-of-the-art algorithms for solving our DRO problem. The sample complexity is measured in terms of finding an $\epsilon$-stationary point w/o PL condition, i.e., $\|\nabla F(\mathbf{w})\|^2 \leq \epsilon$, or achieving $\epsilon$-objective gap, i.e, $F(\mathbf{w}) - \min_{\mathbf{w}} F(\mathbf{w}) \leq \epsilon$ with PL condition. $\widetilde{O}$ omits a logarithmic dependence over $\epsilon$. $n$ represents the size of datasets for a finite sum problem, $d$ denotes the dimension of $\mathbf{w}$. GDS represents whether the step size is geometrically decreased.

| Settings | Algorithms | Sample Complexity | batch size | GDS $\eta$ | Memory Cost | Style |
|---|---|---|---|---|---|---|
| w/o PL | PG-SMD2 [41] | $O(n/\epsilon + 1/\epsilon^2)$ | $O(1)$ | x | $O(n+d)$ | Primal-Dual |
| | ASC-PG [45] | $O(1/\epsilon^2)$ | $O(1)$ | x | $O(d)$ | Compositional |
| | CIVR [53] | $O(1/\epsilon^{3/2})$ | $O(1/\epsilon)$ | x | $O(d)$ | Compositional |
| | **COVER** (This paper) | $\widetilde{O}(1/\epsilon^{3/2})$ | $O(1)$ | x | $O(d)$ | Compositional |
| w/ PL | Stoc-AGDA [50] | $O(1/\mu^2\epsilon)$ | $O(1)$ | x | $O(n+d)$ | Primal-Dual |
| | PES-SGDA [15] | $O(1/\mu^2\epsilon)$ | $O(1)$ | ✓ | $O(n+d)$ | Primal-Dual |
| | RCIVR [53] | $\widetilde{O}(1/\mu\epsilon)$ | $O(1/\epsilon)$ | x | $O(d)$ | Compositional |
| | **RECOVER** (This paper) | $O(1/\mu\epsilon)$ | $O(1)$ | ✓ | $O(d)$ | Compositional |

In addition, this paper also makes several important theoretical contributions for stochastic non-convex optimization, including

1. We establish a nearly optimal complexity for finding $\epsilon$-stationary point, i.e., $\|\nabla F(\mathbf{w})\|^2 \leq \epsilon$, for a class of two-level compositional problems in the order of $\widetilde{O}(1/\epsilon^{3/2})$ without a large mini-batch size, which is better than existing results [44, 45, 14, 5].

2. We etablish an optimal complexity for finding $\epsilon$-optimal solution under an $\mu-$PL condition for a class of two-level compositional problems in the order of $O(1/(\mu\epsilon))$ without a large mini-batch size, which is better than existing results [53].

A theoretical comparison between our results and existing results is shown in Table 1. Empirical studies vividly demonstrate the effectiveness of RECOVER for deep learning on imbalanced data.

## 2  Related Work

DRO has been extensively studied in machine learning [31, 11, 40], statistics, and operations research [42]. In [33], the authors proved that minimizing the DRO formulation with a quadratic regularization in a constraint form is equivalent to minimizing the sum of the empirical loss and a variance regularization defined on itself. Variance regularization can enjoy better generalization error compared with the empirical loss minimization [33], and was also observed to be effective for imbalanced data [33, 56]. Recently, [12] also establishes this equivalence for a broader family of regularization function $h(\mathbf{p}, \mathbf{1}/n)$ including the KL divergence.

Several recent studies have developed stochastic primal-dual methods for solving DRO with a non-convex loss function $\ell(\mathbf{w}; \mathbf{z})$ assuming it is smooth or weakly convex [41, 26, 30, 48]. [41] proposed the first primal-dual methods for solving weakly convex concave problems. For online problems, their algorithms for finding an $\epsilon$-stationary solution whose gradient norm square is less than $\epsilon$ have a sample complexity of $\tilde{O}(1/\epsilon^2)$ or $O(1/\epsilon^3)$ with or without leveraging the strong concavity of $h(\mathbf{p}, \mathbf{1}/n)$ for finding an $\epsilon$-stationary point. Recently Liu et al. [28] proposed to leverage the PL condition of the objective function to improve the convergence for a non-convex min-max formulation of AUC maximization. Then, a PES-SGDA algorithm is proposed to solve a more general class of non-convex min-max problems by leveraging the PL condition [15]. Both [28] and [15] have used geometrically decreasing step sizes in a stagewise manner. However, their algorithms' complexity is in the order of $O(1/\mu^2\epsilon)$, which is worse than $O(1/\mu\epsilon)$ achieved in this paper. Similarly, [50] also leveraged PL conditions to solve non-convex min-max problems and has a sample complexity of $O(1/\mu^2\epsilon)$. Nevertheless, the step size of their algorithm is decreased polynomially in the order of $O(1/t)$, which usually yields poor performance for deep learning.

All the methods reviewed above require maintaining and updating both the primal variable $\mathbf{w}$ and a high dimensional dual variable $\mathbf{p} \in \mathbb{R}^n$. Recently, Levy et al. [23] considered different formulations of DRO, which includes our considered KL-regularized DRO formulation as a special case. Their assumed that the loss function is convex and proposed a stochastic method with a sample complexity $O(1/\epsilon^3)$ for sovling the KL-regularized DRO formulation. In contrast, we provide a better sample

complexity in the order of $O(1/\epsilon)$ under a PL condition without convexity assumption. Additionally, their method requires a large batch size in the order of $O(1/\epsilon)$, while our method only requires a constant batch size which is more practical. We also notice that a recent work [24] and its extended version [25] have considered a formulation similar to (2) and proposed a stochastic algorithm. However, their algorithm has a slower convergence rate with an $O(1/\epsilon^2)$ complexity for finding an $\epsilon$-stationary point and an $O(1/(\mu^2\epsilon))$ complexity for finding an $\epsilon$-optimal solution under a PL condition. Our work is a concurrent work appearing online earlier than [24]. To the best of our knowledge, this is the first work trying to solve the non-convex DRO problem with a duality-free stochastic method by formulating the min-max formulation into an equivalent stochastic compositional problem.

There are extensive studies for solving stochastic compositional problems. [44] considered a more general family of stochastic compositional problems and proposed two algorithms. When the objective function is non-convex, their algorithm's complexity is $O(1/\epsilon^{7/2})$ for finding an $\epsilon$-stationary solution. This complexity was improved in their later works [14], reducing to $O(1/\epsilon^2)$. When the objective is smooth, several papers proposed to use variance reduction techniques (e.g., SPIDER, SARAH) to improve the complexity for finding a stationary point [53, 18, 55, 27]. The best sample complexity achieved for online problems is $O(1/\epsilon^{3/2})$ [53, 55]. [53] also considered the PL condition for developing a faster algorithm called restarted CIVR, whose sample complexity is $O(1/\mu\epsilon)$. However, these variance reduction-based methods require using a very large mini-batch size at many iterations, which has detrimental influence on training deep neural networks [43]. To address this issue, [9] proposed a new technique called STORM that integrates momentum and the recursive variance reduction technique for solving stochastic smooth non-convex optimization. Their algorithm does not require a large mini-batch size at every iterations and enjoys a sample complexity of $O(\log^{2/3}(1/\epsilon)/\epsilon^{3/2})$ for finding an $\epsilon$-stationary point. However, their algorithm uses a polynomially decreasing step size, which is not practical for deep learning, and is not directly applicable to stochastic composite problems.

## 3 Preliminaries

In this section, we provide some definitions and assumptions for next section. For more generality, we consider the stochastic compositional problem (4):

$$\min_{\mathbf{w}\in\mathbb{R}^d} F(\mathbf{w}) := f(\mathbb{E}_{\mathbf{z}}[g_{\mathbf{z}}(\mathbf{w})]) + r(\mathbf{w}) \tag{5}$$

where $g_{\mathbf{z}}(\mathbf{w}) : \mathbb{R}^d \to \mathbb{R}^p$. Define $g(\mathbf{w}) = \mathbb{E}_{\mathbf{z}}[g_{\mathbf{z}}(\mathbf{w})]$. Let $\|\cdot\|$ denote the Euclidean norm of a vector or the Frobenius norm of a matrix. We make the following standard assumptions regarding the problem (5).

**Assumption 1.** *Let $C_f, L_f, C_g$ and $L_g$ be positive constants. Assume that*

*(a) $f : \mathbb{R}^p \to \mathbb{R}$ is a $C_f$-Lipschitz function and its gradient $\nabla f$ is $L_f$-Lipschitz.*

*(b) $g_{\mathbf{z}} : \mathbb{R}^d \to \mathbb{R}^p$ satisfies $\mathbb{E}\|g_{\mathbf{z}}(\mathbf{w}_1) - g_{\mathbf{z}}(\mathbf{w}_2)\|^2] \leq C_g^2\|\mathbf{w}_1 - \mathbf{w}_2\|^2$ for any $\mathbf{w}_1, \mathbf{w}_2$ and its Jacobian $\nabla g_{\mathbf{z}}$ satisfies $\mathbb{E}[\|\nabla g_{\mathbf{z}}(\mathbf{w}_1) - \nabla g_{\mathbf{z}}(\mathbf{w}_2)\|^2] \leq L_g^2\|\mathbf{w}_1 - \mathbf{w}_2\|^2$.*

*(c) $r : R^d \to \mathbb{R} \cup \{\infty\}$ is a convex and lower-semicontinuous function.*

*(d) $F_* = \inf_{\mathbf{w}} F(\mathbf{w}) \geq -\infty$ and $F(\mathbf{w}_1) - F_* \leq \Delta_F$ for the initial solution $\mathbf{w}_1$.*

**Remark:** When $f(s) = s$ is a linear function, the assumption $\mathbb{E}\|g_{\mathbf{z}}(\mathbf{w}_1) - g_{\mathbf{z}}(\mathbf{w}_2)\|^2] \leq C_g^2\|\mathbf{w}_1 - \mathbf{w}_2\|^2$ is not needed. To upper bound continuity and smoothness of function $F$, we denote $L = 2\max\{L_g C_g L_f, C_f C_g L_f, C_f^2, L_g C_f, C_g^2 L_f, C_f, C_g L_f, C_f^2, C_g^2, C_g^2 L_f^2\}$ for simple derivation in the appendix.

**Assumption 2.** *Let $\sigma_g$ and $\sigma_{g'}$ be positive constants and $\sigma^2 = \sigma_g^2 + \sigma_{g'}^2$. Assume that*

$$\mathbb{E}_{\mathbf{z}}[\|g_{\mathbf{z}}(\mathbf{w}) - g(\mathbf{w})\|^2] \leq \sigma_g^2, \; \mathbb{E}_{\mathbf{z}}[\|\nabla g_{\mathbf{z}}(\mathbf{w}) - \nabla g(\mathbf{w})\|^2] \leq \sigma_{g'}^2.$$

**Remark:** We remark how the minimization formulation of DRO problem (3) can satisfy Assumption 1, in particular Assumption 1(a) and (b). In order to satisfy Assumption 1(b), we can define a bounded loss function $\ell(\mathbf{w}, \mathbf{z}) \in [0, \ell_{\max}]$ and then use a shifted loss $\ell(\mathbf{w}; \mathbf{z}) - \ell_{\max}$ in (3). Then $g_{\mathbf{z}}(\mathbf{w}) = \exp((\ell(\mathbf{w}; \mathbf{z}) - \ell_{\max})/\lambda)$ is Lipschitz continuous and smooth if $\ell(\mathbf{w}; \mathbf{z})$ is Lipschitz and smooth. $f(s) = \lambda \log(s)$ is Lipschitz continuous and smooth since $s \geq \exp(-\ell_{\max}/\lambda)$.

For more generality, we allow for a non-smooth regularizer $r(\cdot)$ in this section. To handle non-smoothness of $r$, we can use the proximal operator of $r$: $\mathbf{prox}_r^\eta(\bar{\mathbf{w}}) = \arg\min_{\mathbf{w}} \frac{1}{2}\|\mathbf{w} - \bar{\mathbf{w}}\|^2 + \eta r(\mathbf{w})$. When $r = 0$, the above operator reduces to the standard Euclidean projection. Correspondingly, we define the proximal gradient measure for the compositional problem (5):

$$\mathcal{G}_\eta(\mathbf{w}) = \frac{1}{\eta}(\mathbf{w} - \mathbf{prox}_r^\eta(\mathbf{w} - \eta\nabla g(\mathbf{w})^\top \nabla f(g(\mathbf{w})))).$$

When $r = 0$, the proximal gradient reduces to the standard gradient measure, i.e., $\mathcal{G}_\eta(\mathbf{w}) = \nabla F(\mathbf{w})$. To facilitate our discussion, we define sample complexity below.

**Definition 1.** *The **sample complexity** is defined as the number of samples $\mathbf{z}$ in order to achieve $\mathbb{E}[\|\mathcal{G}_\eta(\mathbf{w})\|^2] \leq \epsilon$ for a certain $\eta > 0$ or $\mathbb{E}[F(\mathbf{w}) - F_*] \leq \epsilon$.*

# 4 Basic Algorithm: COVER

We present our Algorithm 1,which serves as the foundation for proving the the convergence of the objective gap under a PL condition in next section. The convergence results in this section might be of independent interest to those who are interested in convergence analysis without a PL condition. The motivation is to develop a stochastic algorithm with fast convergence in terms of gradient norm. We refer to the algorithm as Compositional Optimal VariancE Reduction (COVER). It will be clear shortly why it is called optimal variance reduction. Note that in order to compute a stochastic estimator of the gradient $f(g(\mathbf{w}))$ given by $\nabla g(\mathbf{w})^\top \nabla f(g(\mathbf{w}))$, we maintain and update two estimators denoted by $\{\mathbf{u}\}_{t=1}^T$ and $\{\mathbf{v}\}_{t=1}^T$ sequence, respectively. The $\{\mathbf{u}_t\}_{t=1}^T$ sequence maintains an estimation of $\{g(\mathbf{w}_t)\}_{t=1}^T$ and the $\{\mathbf{v}_t\}_{t=1}^T$ sequence maintains an estimation of $\{\nabla g(\mathbf{w}_t)\}_{t=1}^T$. The strategy of maintaining and updating two individual sequences was first proposed in [44] and has been widely used for solving compositional problems [55, 53]. However, the key difference from previous algorithms lies in the method for updating the two sequences. COVER is inspired by the STORM technique [9]. To understand the update, let us consider update that applied to the DRO problem (3) by let $f(\cdot) = \lambda\log(\cdot)$, $g_{\mathbf{z}}(\cdot) = \exp(\frac{\ell(\cdot;\mathbf{z})}{\lambda})$ and ignoring $r$ for the moment. Plugging the gradient of $f(\cdot)$ and $g_{\mathbf{z}}(\cdot)$, we have

$$\mathbf{w}_{t+1} = \mathbf{w}_t - \eta_t \frac{1}{u_t}\widetilde{\mathbf{v}}_t,$$

$$\widetilde{\mathbf{v}}_{t+1} = \exp(\frac{\ell(\mathbf{w}_{t+1};\mathbf{z}_{t+1})}{\lambda})\nabla\ell(\mathbf{w}_{t+1};\mathbf{z}_{t+1}) + (1 - a_{t+1})(\widetilde{\mathbf{v}}_t - \exp(\frac{\ell(\mathbf{w}_t;\mathbf{z}_{t+1})}{\lambda})\nabla\ell(\mathbf{w}_t;\mathbf{z}_{t+1})),$$

where $u_t$ becomes a scalar, which is an online variance-reduced estimator of $\mathbb{E}_{\mathbf{z}}[\exp(\ell(\mathbf{w}_t;\mathbf{z})/\lambda)]$, and $\widetilde{\mathbf{v}}_t$ is a scaled version of $\mathbf{v}_t$, which is an online variance-reduced estimator of $\mathbb{E}_{\mathbf{z}}[\exp(\ell(\mathbf{w}_t;\mathbf{z})/\lambda)\nabla\ell(\mathbf{w}_t;\mathbf{z})]$.

Finally, we notice that a similar method for updating the $\mathbf{u}_t$ sequence for estimating $g(\mathbf{w}_t)$ has been adopted in a recent work [6]. However, different from the present work they just use an unbiased stochastic gradient to estimate $\nabla g(\mathbf{w}_t)$, which yields a worse convergence rate.

## 4.1 Convergence of Proximal Gradient

In this section, we present the convergence result of COVER.

**Theorem 1.** *Assume the Assumption 1 and 2, for any $C > 0$, $k = \frac{C\sigma^{2/3}}{L}$, $c = 128L + \sigma^2/(7Lk^3)$, $w = \max((16Lk^3), 2\sigma^2, (\frac{ck}{4L})^3)$, and $\eta_t = k/(w + \sigma^2 t)^{1/3}$. The output of COVER satisfies*

$$\mathbb{E}[\|\mathcal{G}_{\eta_{t*}}(\mathbf{w}_{t_*})\|^2] \leq \widetilde{O}\left(\frac{\Delta_F}{T^{2/3}} + \frac{\sigma^2}{T^{2/3}}\right). \tag{6}$$

*where $t_*$ is sampled from $\{1, \ldots, T\}$.*

**Remark:** Theorem 1 implies that with a polynomially decreasing step size, COVER is able to find an $\epsilon$-stationary point, i.e., $\mathbb{E}[\|\mathcal{G}_{\eta_{t*}}(\mathbf{w}_{t_*})\|^2] \leq \epsilon$ for a regularized objective and $\mathbb{E}[\|\nabla F(\mathbf{w})\|^2] \leq \epsilon$ for a non-regularized objective, with a near-optimal sample complexity $\widetilde{O}(\frac{1}{\epsilon^{3/2}})$. Note that the complexity $\widetilde{O}(1/\epsilon^{3/2})$ is optimal up to a logarithmic factor for making the (proximal) gradient's norm smaller than $\epsilon$ in expectation for solving non-convex smooth optimization problems [2].

**Algorithm 1:** COVER $(\mathbf{w}_1, \mathbf{u}_1, \mathbf{v}_1, \{\eta_t\}, T, \text{PL} = \text{False})$

1: Let $a_t = c\eta_t^2$
2: **if** not PL **then**
3:     Draw a samples $\mathbf{z}$ and construct the estimates: $\mathbf{u}_1 = g_{\mathbf{z}}(\mathbf{w}_1)$, $\mathbf{v}_1 = \nabla g_{\mathbf{z}}(\mathbf{w}_1)$
4: **end if**
5: **for** $t = 1, \ldots, T - 1$ **do**
6:     $\mathbf{w}_{t+1} \leftarrow \mathbf{prox}_r^{\eta_t}(\mathbf{w}_t - \eta_t \mathbf{v}_t^\top \nabla f(\mathbf{u}_t))$
7:     Draw a samples $\mathbf{z}_{t+1}$, and update

$$\mathbf{u}_{t+1} = g_{\mathbf{z}_{t+1}}(\mathbf{w}_{t+1}) + (1 - a_{t+1})(\mathbf{u}_t - g_{\mathbf{z}_{t+1}}(\mathbf{w}_t))$$

$$\mathbf{v}_{t+1} = \nabla g_{\mathbf{z}_{t+1}}(\mathbf{w}_{t+1}) + (1 - a_{t+1})(\mathbf{v}_t - \nabla g_{\mathbf{z}_{t+1}}(\mathbf{w}_t))$$

8: **end for**
9: **Return:** $(\mathbf{w}_\tau, \mathbf{u}_\tau, \mathbf{v}_\tau)$ for randomly selected $\tau \in \{1, \ldots, T\}$.

## 5 A Practical Variant (RECOVER) under a PL condition

The issue of COVER is that the polynomially decreasing step size is not practical for deep learning applications and obstacles its generalization performance [52]. A stagewise step size is widely and commonly used [17, 22, 52] for deep learning optimization. To this end, we develop a multi-stage REstarted version of COVER, called RECOVER, which uses a geometrically decreasing step size in a stagewise manner. In oder to analyze RECOVER, we assume the following PL condition of the objective with a smooth regularization $r$ term [52].

**Assumption 3.** $F(\mathbf{w})$ *satisfies the* $\mu$-*PL condition if there exists* $\mu > 0$ *such that*

$$2\mu(F(\mathbf{w}) - \min_{\mathbf{w} \in \mathbb{R}^d} F(\mathbf{w})) \le \|\nabla F(\mathbf{w})\|^2. \tag{7}$$

In the following, we simply consider the objective $F(\mathbf{w}) = f(\mathbb{E}_{\mathbf{z}}[g_{\mathbf{z}}(\mathbf{w})])$, where $r(\cdot)$ is absorbed into $f(\mathbb{E}_{\mathbf{z}}[g_{\mathbf{z}}(\mathbf{w})])$. As a result, $\mathcal{G}_\eta(\mathbf{w}) = \nabla F(\mathbf{w})$.

Although the PL condition has been considered in various papers for developing stagewise algorithms and improving the convergence rate of non-convex optimization [53, 52, 28, 15]. In order to establish the improved rate, we have innovations in twofold (i) at the algorithmic level, we utilize the variance reduction techniques at the inner and outer level without using mega large mini-batch size at any iterations; (ii) at the analysis level, we innovatively prove that the estimation error of the two sequences, $\mathbf{u}$ and $\mathbf{v}$, are decreasing geometrically after a stage (Lemma 3). These innovations at two levels yield the optimal convergence rate in the order of $O(1/(\mu\epsilon))$.

### 5.1 Theoretical Verification of PL Assumption for KL-regularized DRO

Before presenting the proposed algorithm and its convergence, we discuss how the $F_{dro}$ can satisfy Assumption 3. First, we note that a PL condition of the weighted loss implies that of the primal objective.

**Lemma 1.** *Let* $F_{\mathbf{p}}(\mathbf{w}) = \sum_{i=1}^n p_i \ell(\mathbf{w}; \mathbf{z}_i)$. *If for any* $\mathbf{p} \in \Delta_n$, $F_{\mathbf{p}}(\mathbf{w})$ *satisfies a* $\mu$-*PL condition, then* $F_{dro}(\mathbf{w}) = \lambda \log(\frac{1}{n} \sum_i \exp(\ell(\mathbf{w}; \mathbf{z}_i)/\lambda))$ *satisfies the* $\mu$-*PL condition.*

**Remark:** The assumption that the weighted loss satisfies a PL condition can be proven for a simple square loss $\ell(\mathbf{w}; \mathbf{z}_i) = (\mathbf{w}^\top \mathbf{x}_i - y_i)^2$, where $\mathbf{z}_i = (\mathbf{x}_i, y_i)$ consists of a feature vector $\mathbf{x}_i$ and a label $y_i$. In order to see this, we can write $F_p(\mathbf{w}) = \sum_{i=1}^n (\mathbf{w}^\top \mathbf{x}_i \sqrt{p_i} - y_i \sqrt{p_i})^2 = \|A\mathbf{w} - \mathbf{b}\|^2$, where $A = (\mathbf{x}_1 \sqrt{p_1}, \ldots, \mathbf{x}_n \sqrt{p_n})^\top$, $\mathbf{b} = (y_1 \sqrt{p_1}, \ldots, y_n \sqrt{p_n})^\top$. It has been shown in many previous studies that such $F_{\mathbf{p}}(\mathbf{w})$ satisfies a PL condition [47, 51, 34]. Hence, the above lemma indicates $F_{dro}(\mathbf{w})$ satisfies a PL condition.

We can also justify that $F_{dro}(\mathbf{w})$ satisfies a PL condition for deep learning with ReLU activation function in a neighborhood around a random initialized point following the result in [1].

**Lemma 2.** *Assume that input* $\{(\mathbf{x}_1, y_1), \ldots, (\mathbf{x}_n, y_n)\}$ *satisfies* $\|\mathbf{x}_i\| = 1$ *and* $\|\mathbf{x}_i - \mathbf{x}_j\| \ge \delta$, *where* $\mathbf{x}_n \in \mathbb{R}^{d_1}$, $y_i \in \mathbb{R}^{d_0}$ *and* $\|y_i\| \le O(1)$. *Consider a deep neural network with* $h_{i,0} = \phi(A\mathbf{x}_i)$, $h_{i,l} =$

**Algorithm 2:** RECOVER($\mathbf{w}_0, \epsilon_0, c$)

1: **Initialization**: Draw a sample $\mathbf{z}_0$ and construct the estimates $\mathbf{u}_0 = g_{\mathbf{z}_0}(\mathbf{w}_0)$, $\mathbf{v}_0 = \nabla g_{\mathbf{z}_0}(\mathbf{w}_0)$
2: **for** $k = 1, \ldots, K$ **do**
3: $\quad (\mathbf{w}_k, \mathbf{u}_k, \mathbf{v}_k) = \text{COVER}(\mathbf{w}_{k-1}, \mathbf{u}_{k-1}, \mathbf{v}_{k-1}, \eta_k, T_k, \text{True})$
4: $\quad$ change $\eta_k, T_k$ according to Theorem 2
5: **end for**
6: **Return:** $\mathbf{w}_K$

---

$\phi(W_l h_{i,l-1}), l = 1, \ldots, \tilde{L}, \hat{y}_i = Bh_{i,\tilde{L}}$ where $A \in \mathbb{R}^{d_2 \times d_1}$ $W_l \in \mathbb{R}^{d_2 \times d_2}$, $B \in \mathbb{R}^{d_0 \times d_2}$, $\phi$ is the ReLU activation function, and $\ell(W; \mathbf{z}_i) = (\hat{y}_i - y_i)^2$ is a square loss. Suppose that for any $W$, $p_i^* = \exp(\ell(W; \mathbf{z}_i)/\lambda) / \sum_{i=1}^n \exp(\ell(W; \mathbf{z}_i)/\lambda) \geq p_0 > 0$, then with a high probability over randomness of $W_0, A, B$ for every $W$ with $\|W - W_0\| \leq O(1/poly(n, \tilde{L}, p_0^{-1}, \delta^{-1}))$, there exists a small $\mu > 0$ such that $\|\nabla F_{dro}(W)\|_F^2 + O(\epsilon) \geq \mu(F_{dro}(W) - \min_W F_{dro}(W))$.

**Remark:** The $O(\epsilon)$ term in the left side of the PL condition is caused by using the covering net argument for proving the high probability result. Nevertheless, it does not affect the final convergence rate.

### 5.2 Theoretical Analysis of RECOVER

Now, we are ready to present the proposed algorithm under the PL condition and its convergence result. The algorithm is described in Algorithm 2.

The first key feature of RECOVER is equipped with the practical geometrical decreases step size between stages. At each stage, we adopt a constant step size $\eta_k$ rather than the polynomial decreases step size used by COVER as in Theorem 1. Another key feature of RECOVER is that it uses not only $\mathbf{w}_k$ for restarting but also $\mathbf{u}_k, \mathbf{v}_k$, the corresponding online estimator of $g(\mathbf{w}_k)$ and $\nabla g(\mathbf{w}_k)$, for restarting the next stage. It is this feature that allows us to avoid the large batch size required in other variance reduction methods to achieve the optimal sample complexity. With this feature, we can show that the variance of $\mathbf{u}_k, \mathbf{v}_k$ is decreased by a constant factor stagewisely as shown in the following lemma.

**Lemma 3.** *Define constants $\epsilon_1 = \frac{c^2 \sigma^2}{64 \mu L^3}$ and $\epsilon_k = \epsilon_1/2^{k-1}$, with $\eta_k = \min\{\frac{\sqrt{\mu \epsilon_k} L}{2c\sigma}, \frac{1}{16L}\}$, $T_k = O(\max\{\frac{96 c\sigma}{\mu^{3/2} \sqrt{\epsilon_k} L}, \frac{16 c^2 \sigma^2}{\mu L^2 \epsilon_k}, \frac{\Delta_F}{\sigma^2}\})$, the variance of the stochastic estimator of Algorithm 2 at $\mathbf{w}_k$ satisfies:*

$$\mathbb{E}[\|\mathbf{u}_k - g(\mathbf{w}_k))\|^2 + \|\mathbf{v}_k - \nabla g(\mathbf{w}_k))\|^2] \leq \mu \epsilon_k. \tag{8}$$

With the above lemma and the convergence bound for $\mathbb{E}[\|\nabla F(\mathbf{w}_k)\|^2]$ at the $k$-th stage, we can show that the objective gap $\mathbb{E}[F(\mathbf{w}_k) - F_*]$ is decreased by a factor of 2 after each stage under the PL condition. Hence, we have the following convergence for RECOVER.

**Theorem 2.** *Assume that assumption 1,2,3 hold. Define constants $\epsilon_1 = \frac{c^2 \sigma^2}{64 \mu L^4}$ and $\epsilon_k = \epsilon_1/2^{k-1}$. By setting $\eta_k = \min\{\frac{\sqrt{\mu \epsilon_k} L}{2c\sigma}, \frac{1}{16L}\}$, $T_k = O(\max\{\frac{96 c\sigma}{\mu^{3/2} \sqrt{\epsilon_k} L}, \frac{2 c^2 \sigma^2}{\mu L^2 \epsilon_k}, \frac{\Delta_F}{\sigma^2}\})$, $c = 104 L^2$, then after $K = O(\log(\epsilon_1/\epsilon))$ stages, the output of RECOVER satisfies $\mathbb{E}[F(\mathbf{w}_K) - F_*] \leq \epsilon$.*

**Remark:** It is not difficult to derive the sample complexity of RECOVER is $O(\max\{\frac{1}{\mu^{3/2} \sqrt{\epsilon}}, \frac{1}{\mu \epsilon}\}) = O(\frac{1}{\mu \epsilon})$ for $\epsilon \leq \mu$. It is notable this complexity is optimal for the considered general stochastic compositional problem, which includes stochastic strongly convex optimization as a special case, whose lower bound is $O(1/(\mu \epsilon))$ [16].

In addition, it is notable that the proposed multi-stage algorithm is very different from many other multi-stage algorithms for non-convex optimization that are based on the proximal point framework [7, 48, 41, 15]. In particular, in these previous studies, a quadratic function $\gamma/2\|\mathbf{w} - \mathbf{w}_{k-1}\|^2$ with an appropriate regularization parameter $\gamma$ is added into the objective function at the $k$-th stage in order to convexify the objective function. In RECOVER, no such regularization is manually added. Nevertheless, we can still obtain strong convergence guarantee.

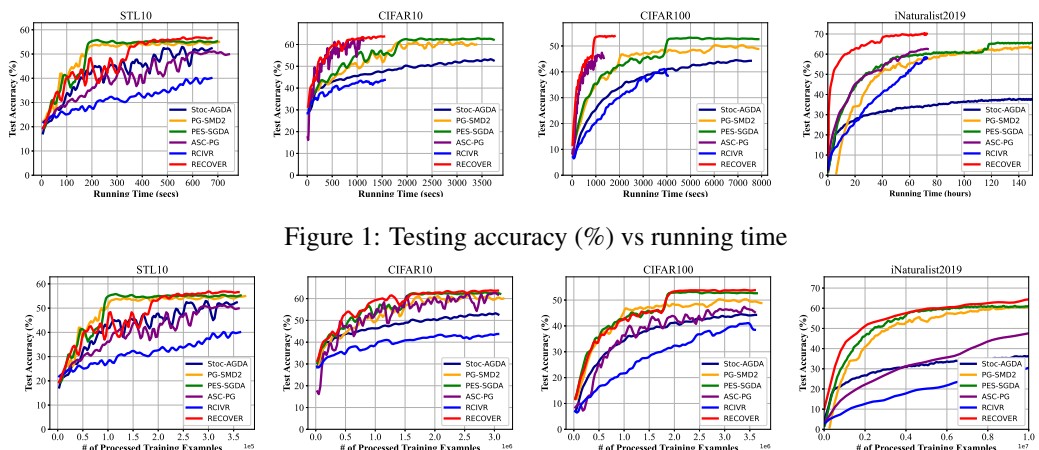

Figure 1: Testing accuracy (%) vs running time

Figure 2: Testing accuracy (%) vs # of processed training examples

## 6 Experimental Results

We focus on the task of classification with imbalanced data in our experiments. Firstly, we compare RECOVER with five State-Of-The-Art (SOTA) baselines from two categories: (i) primal-dual algorithms for solving the primal-dual formulation of DRO (1), and (ii) algorithms that are designed for the stochastic compositional formulation of DRO (3). Secondly, we verify the advantages of DRO over Emperical Risk Minimization (ERM) for imbalanced data problems by comparing the test accuracy learned by optimizing DRO using RECOVER and optimizing ERM using SGD on the imbalanced datasets. Then we show the RECOVER is also an effective fine-tuning algorithm for large-scale imbalanced data training. The code for reproducing the results is released here [39].

### 6.1 Comparison with SOTA DRO Baselines

We compare RECOVER with five baselines: Restarted CIVR [53] (RCIVR), ASC-PG [45], Stoc-AGDA [50], PG-SMD2 [41] and PES-SGDA [15]. RCIVR and ASC-PG are the state-of-the-art algorithms for solving stochastic compositional problems. RCVIR uses variance reduction techniques and leverages the PL condition, while ASC-PG does neither. Stoc-AGDA and PG-SMD2 are the primal-dual algorithms with and without leveraging the PL condition explicitly, respectively. PES-SGDA is a variant of PG-SMD2 and was proposed by leveraging the PL condition for achieving faster convergence. Please note that ASC-PG and Stoc-AGDA use polynomially decreasing step sizes, RECOVER, PG-SMD2 and PES-SGDA use stagewise decreasing step size, and RCIVR uses a constant step size. The parameters of each algorithm are appropriately tuned for the best performance. All the algorithms are implemented using Pytorch and run on GeForce GTX 1080 Ti GPU.

We conduct experiments on four datasets, namely STL10 [8], CIFAR10, CIFAR100 [21], and iNaturalist2019 [46]. The original STL10, and CIFAR10, CIFAR100 are balanced data, where STL10 has 10 classes and each class has 500 training images, CIFAR10 (resp. CIFAR100) has 10 (resp. 100) classes and each class has 5K (resp. 500) training images. For STL10, CIFAR10 and CIFAR100, we artificially construct imbalanced training data, where we only keep the last 100 images of each class for the first half classes. iNaturallist2019 itself is an imbalanced dataset that contains 265,213 images with 1010 classes. We train ResNet-20 on STL10, CIFAR10, CIFAR100, and Inception-V3 on iNaturalist2019.

For fair comparison, we use the same constant batch size $b$ for all methods except for RCIVR in which the inner loop batch size $b'$ and outer loop batch size $B_k$ are hyperparameters that relate to convergence. We use the constant batch size $b = 128$ on CIFAR10, CIFAR100, and $b = 64$ on iNaturalist2019, and $b = 32$ on STL. For RCIVR, both the fixed inner loop batch size $b'$ and the initial outer loop batch size $B_0$ are tuned in $\{32, 64, 128\}$. The outer loop mini-batch size $B_k$ is also increased by a factor of 10 per-stage according to the theory.

Table 2: Test accuracy (%), mean (variance), of SGD for ERM and RECOVER for DRO. Bold numbers represent better performance.

| IMRATIO | STL10 | | CIFAR10 | | CIFAR100 | |
|---|---|---|---|---|---|---|
| | SGD | RECOVER | SGD | RECOVER | SGD | RECOVER |
| 0.02 | 37.97 (0.78) | **38.08 (0.59)** | 65.36(0.64) | **66.14 (0.48)** | 38.99 (0.62) | **39.45 (0.56)** |
| 0.05 | 41.12 (0.94) | **42.68 (0.60)** | 74.74 (0.71) | **75.90 (0.33)** | 45.79 (0.69) | **46.47 (0.66)** |
| 0.1 | 46.03 (0.96) | **48.94 (0.86)** | 79.32 (0.42) | **80.93 (0.31)** | 49.45 (0.50) | **50.84 (0.86)** |
| 0.2 | 51.75 (1.14) | **56.06 (1.26)** | 84.84 (0.51) | **85.93 (0.14)** | 55.80 (0.74) | **56.90 (0.42)** |

| Model | ImageNet-LT | Places-LT |
|---|---|---|
| CE (SGD) | 41.29 (3e-3) | 27.47 (1e-3) |
| Focal (SGD) | 41.10 (2e-2) | 27.64 (6e-3) |
| DRO (RECOVER) | **42.30** (4e-4) | **28.75** (4e-5) |

Figure 3: Left: Test Accuracy vs $\lambda$ on CIFAR10 data; Right: Test accuracy (%) of finetuned models by different methods.

For RECOVER, the initial step size $\eta_0$ and the momentum parameter $a_0$ at the first stage are tuned in $\{0.1, 0.2, ..., 1\}$, and $\eta_k$ is divided by 10 after each stage and $a_k$ is updated accordingly. For RCIVR, the constant step size is tuned $\eta \in \{0.1, 0.2, \cdots, 1\}$. For the ASC-PG, the step size is set to be $\eta = c_0/t^a$, and the momentum parameter is set to be $\beta = 2c_0/t^b$, where $c_0$ is tuned from $0.01 \sim 1$ and $a, b$ are tuned ranging from 0.1 to 0.9 by grid search, $t$ is the number of iterations. For Stoc-AGDA, the step size for primal variable is set to be $\beta_1/(\tau_1 + t)$ and the step size for dual variable $\mathbf{p}$ is set to be $\beta_2/(\tau_2 + t)$. $\beta_1, \beta_2$ are tuned in $[10^{-1}, 1, 10, 10^2, 500, 10^3]$ and $\tau_1, \tau_2$ are tuned in $[1, 10, 10^2, 500, 10^3]$. For PES-SGDA and PG-SMD2, the algorithm have multiple stages, where each stage solves a strongly-convex strongly-concave subproblem, and step size decrease after each stage. For PES-SGDA, the number of iteration per-stage is increased by a factor of 10 and step sizes for the primal and the dual variables are decreased by 10 times after each stage, with their initial values tuned. In particular, $\eta_1$ (for primal variable) is tuned in $\{0.1, 0.2, \cdots 1\}$ and $\eta_2$ (for the dual variable) is tuned in $\{10^{-5}, 10^{-4}, 10^{-3}\}$, $T_0$ (the number of iterations for the first stage) is tuned in $\{5, 10, 30, 60\}\frac{n}{b}$, where $n$ is the number of training examples.

As we aim to compare the optimization for the same objective in this section, $\lambda$ is set to 5 both in the compositional objective (3) and min-max formulation of (1) with regularizer $h(\mathbf{p}, \mathbf{1}/n) = \lambda \sum_i p_i \log(np_i)$. Following the standard training strategy, we run all algorithms 120 epochs and set the time threshold 150 hours for early stopping on iNaturalist data.

We compare testing accuracy vs running time and vs the number of processed training examples separately. We present the convergence of testing accuracy in terms of running time in Figure 1 and in terms of processed training examples in Figure 2. From the results, we can observe that: (i) in terms of running time RECOVER converges faster than all baselines on all data except on the smallest data STL10, on which PES-SGDA has similar running time performance as RECOVER. The reason is that STL10 is the smallest data, which only has 3000 imbalanced training data samples and hence PES-SGDA has marginal overhead per-iteration; (ii) when the training data size is moderately large, the primal dual methods (PES-SGDA, PG-SMD2, Stoc-AGDA) have significant overhead, which makes them converge much slower than RECOVER in terms of running time. On the large iNaturalist2019 data, RECOVER can save days of training time; (iii) RECOVER is much faster than RCIVR on all datasets; (iv) ASC-PG performs reasonably well but is still not as good as RECOVER in terms of both running time and sample complexity. The convergence instability of ASC-PG verifies the robustness of RECOVER for addressing the compositional problems.

## 6.2 Comparison between SGD and DRO.

We compare the generalization performance of DRO optimized by RECOVER with traditional ERM optimized by SGD for imbalance multi-classification tasks on STL10, CIFAR10, CIFAR100. The IMbalance RATIO (IMRATIO) is defined as the number of samples in the minority classes over

the number of samples in the majority classes. We mannually construct different training sets with different IMRATIO, i.e., we only keep the last IMRATIO portion of images in the first half of classes.

Different from previous experiments, we tune $\lambda$ in a certain of range $\{1, 5, 10, 20, 100\}$ by a cross-validation approach and report the best testing results. Other parameters of RECOVER is tuned according to the setting in previous experiments. We use ResNet-32 for CIFAR10, CIFAR100, and ResNet-20 for STL10. For SGD, the step size is set as $\eta_0$ in the first 60 epochs, and is decreased by a factor of 10 at 60, and 90 epochs following the practical strategy [17], where $\eta_0$ is tuned in $\{0.1, 0.5, 1\}$ and 1 epoch means one pass of training data.

We report averaged test accuracy over 5 runs with mean (variance) in Table 2. We can see that DRO with RECOVER achieves higher test accuracy with smaller variance over multiple runs on all datasets than ERM with SGD. In addition, we report the results over 5 runs of different $\lambda$ on CIFAR10 with different IMRATIO in Figure 3 (left). It is obvious to see that an appropriate regularization on the dual variable can improve the performance.

### 6.3 Effectiveness of RECOVER as a Fine-tuning Method

Fine-tuning high level layers from a pertained model is widely used for transfer learning and is also an effective method to update the models without increasing the computational cost too much when receiving new samples. For this purpose, we demonstrate that DRO is a better objective than the Cross Entropy (CE) loss and focal loss for fine-tuning on imbalanced datasets.

ImageNet-LT [29] and Places-LT [29] are two popular imbalanced data sets and are the Long-Tailed (LT) version of ImageNet-2012 [10] and Places-2 [54] by sampling a subset following the Pareto distribution [3] with the power value 6. ImageNet-LT has 115.8K images from 1000 categories, and Places-LT contains 62.5K training images from 365 classes. The head class is 4980 images and the tail class contains 5 images in both datasets.

To verify that DRO is a better objective and that RECOVER is an efficient optimization algorithm, we compare the test accuracy of the model trained with different objectives: DRO, CE loss and focal loss, where DRO is optimized by RECOVER and the other two losses are optimized by SGD. All methods start from the same pretrained model. We apply the ImangeNet pretrained ResNet152 as the pre-trained model for Places-LT. For ImageNet-LT, we train ResNet50 using CE loss for 90 epochs following the standard training strategy proposed in [17] as the pre-trained model. We then fine tune the last block of the convolutions layer and the classifier layer for 30 epochs by using RECOVER for optimizing DRO and using SGD for optimizing ERM, respectively. The initial step size for RECOVER and SGD are both tuned in $\eta_0 \in \{0.1, 0.5, 1\}$. For DRO, $\lambda$ is tunes in $\{1, 5, 10\}$.

The test accuracy over 3 runs with mean (variance) is reported in Figure 3 (right). It is clear to see that DRO optimized by RECOVER outperforms ERM with the CE loss and focal loss optimized by SGD more than 1(%) on both datasets. This vividly verifies the effectiveness of RECOVER as a fine-tuning method on imbalanced data.

## 7   Conclusion

In this paper, we proposed a duality-free online method for solving a class of distributionally robust optimization problems. We used a KL divergence regularization on the dual variable and transformed the problem into a two-level stochastic compositional problem. By leveraging a practical PL condition, we developed a practical method RECOVER based on recursive variance-reduced estimators and established an optimal sample complexity. Experiments verify the effectiveness of the proposed algorithm in terms of both running time and prediction performance on large-scale imbalanced data. An open question remains is how to solve the DRO problem with a KL constraint on the dual variable by a pratical stochatic algorithm without maintaining and updating the high dimensional dual variable. We plan to address this challenge in the future work.

## Acknowledgments

The authors thank anonymous reviewers for constructive comments. This work was supported by NSF Career Award #1844403, NSF Award #2110545 and NSF Award #1933212.

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
