)$, where $\mathbf{u}_t$, $\mathbf{w}_t$ are the two estimator sequences maintained in COVER. The compositional stochastic variance introduced by $\mathbf{d}_t$ as $\varepsilon_t = \mathbf{d}_t - \nabla g(\mathbf{w}_t)^\top \nabla f(g(\mathbf{w}_t))$, the stochastic variance introduced by $\mathbf{u}_t$ denoted as $\varepsilon_{\mathbf{u}_t} = \mathbf{u}_t - g(\mathbf{w}_t)$, the stochastic variance introduced by $\mathbf{v}_t$ denoted as $\varepsilon_{\mathbf{v}_t} = \mathbf{v}_t - \nabla g(\mathbf{w}_t)$. The stochastic proximal gradient measure of COVER is $\tilde{\mathcal{G}}_\eta(\mathbf{w}_t) = \frac{1}{\eta}(\mathbf{w}_{t+1} - \mathbf{w}_t)$. And $L = 2\max\{L_g C_g L_f, C_f C_g L_f, C_f^2, L_g C_f, C_g^2 L_f, C_f, C_g L_f, C_f^2, C_g^2, C_g^2 L_f^2\}$.

# 8 Illustration of Variance Introduced by $\mathbf{p} \in \mathbb{R}^n$

To see this, the variance of stochastic gradient in terms of $\mathbf{w}$ with random sampling is given by $\text{Var}_r = 1/n \sum_{i=1}^n \|np_i \nabla \ell(\mathbf{w}; \mathbf{z}_i) - \nabla_\mathbf{w} L(\mathbf{w}, \mathbf{p})\|^2 = \sum_{i=1}^n np_i^2 \|\nabla \ell(\mathbf{w}; \mathbf{z}_i)\|^2 - \|\nabla L(\mathbf{w}, \mathbf{p})\|^2$, where $L(\mathbf{w}, \mathbf{p}) = \sum_{i=1}^n p_i \ell(\mathbf{w}; \mathbf{z}_i)$. In contrast, the variance of stochastic gradient in terms of $\mathbf{w}$ with non-uniform sampling according to $\mathbf{p}$ is given by $\text{Var}_n = \sum_{i=1}^n p_i \|\nabla \ell(\mathbf{w}; \mathbf{z}_i) - \nabla_\mathbf{w} L(\mathbf{w}, \mathbf{p})\|^2 = \sum_{i=1}^n p_i \|\nabla \ell(\mathbf{w}; \mathbf{z}_i)\|^2 - \|\nabla L(\mathbf{w}, \mathbf{p})\|^2$. Let us consider an extreme case when $p_i = 1, p_j = 0, \forall j \neq i$, we have $\text{Var}_r = (n-1)\|\nabla \ell(\mathbf{w}; \mathbf{z}_i)\|^2 \gg \text{Var}_n = 0$.

# 9 Proof of Section 4

**Lemma 4.** *Suppose Assumption 1 and 2 hold, we have*

$$\mathbb{E}[\|\varepsilon_t\|^2] \leq 2C_f^2 \mathbb{E}[\|\varepsilon_{\mathbf{v}_t}\|^2] + 2C_g^2 L_f^2 \mathbb{E}[\|\varepsilon_{\mathbf{u}_t}\|^2]. \tag{9}$$

**Remark:** Plugging the definition of $L$ into it, we have $\mathbb{E}[\|\varepsilon_t\|^2] \leq L\mathbb{E}[\|\varepsilon_{\mathbf{v}_t}\|^2] + L\mathbb{E}[\|\varepsilon_{\mathbf{u}_t}\|^2]$

*Proof.*

$$
\begin{aligned}
&\mathbb{E}[\|\mathbf{d}_t - \nabla g(\mathbf{w}_t)^\top \nabla f(g(\mathbf{w}_t))\|^2] = \mathbb{E}[\|\mathbf{v}_t^\top \nabla f(\mathbf{u}_t) - \nabla g(\mathbf{w}_t)^\top \nabla f(g(\mathbf{w}_t)))\|^2]\\
=&\mathbb{E}[\|\mathbf{v}_t^\top \nabla f(\mathbf{u}_t) - \nabla g(\mathbf{w}_t)^\top \nabla f(\mathbf{u}_t) + \nabla g(\mathbf{w}_t)^\top \nabla f(\mathbf{u}_t) - \nabla g(\mathbf{w}_t)^\top \nabla f(g(\mathbf{w}_t)))\|^2]\\
\leq&2\mathbb{E}[\|\mathbf{v}_t^\top \nabla f(\mathbf{u}_t) - \nabla g(\mathbf{w}_t)^\top \nabla f(\mathbf{u}_t)\|^2] + 2\mathbb{E}[\|\nabla g(\mathbf{w}_t)^\top \nabla f(\mathbf{u}_t) - \nabla g(\mathbf{w}_t)^\top \nabla f(g(\mathbf{w}_t))\|^2]\\
\leq&2C_f^2 \mathbb{E}[\|\mathbf{v}_t^\top - \nabla g(\mathbf{w}_t)^\top\|^2] + 2C_g^2 \mathbb{E}[\|\nabla f(\mathbf{u}_t) - \nabla f(g(\mathbf{w}_t))\|^2]\\
\leq&2C_f^2 \mathbb{E}[\|\mathbf{v}_t^\top - \nabla g(\mathbf{w}_t)^\top\|^2] + 2C_g^2 L_f^2 \mathbb{E}[\|\mathbf{u}_t - g(\mathbf{w}_t)\|^2]\\
=&2C_f^2 \mathbb{E}[\|\varepsilon_{\mathbf{v}_t}\|^2] + 2C_g^2 L_f^2 \mathbb{E}[\|\varepsilon_{\mathbf{u}_t}\|^2],
\end{aligned}
$$
$$\tag{10}$$

where the first inequality is due to $\|a + b\|^2 \leq 2\|a\|^2 + 2\|b\|^2$, the second inequality is due to the $C_f$-Lipschitz continuous of $f$, *i.e.*, $\|\nabla f(\mathbf{w}_t)\|^2 \leq C_f^2$, and $C_g$-Lipschitz continuous of $g$, *i.e.*, $\|\nabla g(\mathbf{w}_t)\|^2 \leq C_g^2$. The third inequality is due to the $L_f$-smoothness of $f$ function. $\qquad\square$

**Lemma 5.** *For the two gradient mappings $\|\mathcal{G}_\eta(\mathbf{w}_t)\|^2$, $\|\tilde{\mathcal{G}}_\eta(\mathbf{w}_t)\|^2$, we have*

$$
\begin{aligned}
\mathbb{E}[\|\mathcal{G}_\eta(\mathbf{w}_t)\|^2] &\leq 2\mathbb{E}[\|\tilde{\mathcal{G}}_\eta(\mathbf{w}_t)\|^2] + 2\mathbb{E}[\|\nabla g(\mathbf{w}_t)^\top \nabla f(g(\mathbf{w}_t)) - \mathbf{d}_t\|^2],\\
\mathbb{E}[\|\tilde{\mathcal{G}}_\eta(\mathbf{w}_t)\|^2] &\leq 2\mathbb{E}[\|\mathcal{G}_\eta(\mathbf{w}_t)\|^2] + 2\mathbb{E}[\|\nabla g(\mathbf{w}_t)^\top \nabla f(g(\mathbf{w}_t)) - \mathbf{d}_t\|^2].
\end{aligned}
\tag{11}
$$

**Remark:** This lemma implies that

$$\mathbb{E}[\|\mathbf{w}_{t+1} - \mathbf{w}_t\|^2] = \eta^2 \mathbb{E}[\|\tilde{\mathcal{G}}_\eta(\mathbf{w}_t)\|^2] \leq 2\eta^2 \mathbb{E}[\|\mathcal{G}_\eta(\mathbf{w}_t)\|^2] + 2\eta^2 \mathbb{E}[\|\nabla g(\mathbf{w}_t)^\top \nabla f(g(\mathbf{w}_t)) - \mathbf{d}_t\|^2]. \tag{12}$$

*Proof.* Denote that $\tilde{\mathbf{w}}_{t+1} = \mathbf{prox}_r^\eta(\mathbf{w}_t - \eta \nabla g(\mathbf{w}_t)^\top \nabla f(g(\mathbf{w}_t)))$. Then we have $\|\mathbf{w}_t - \tilde{\mathbf{w}}_{t+1}\|^2 \leq 2\|\mathbf{w}_t - \mathbf{w}_{t+1}\|^2 + 2\|\mathbf{w}_{t+1} - \tilde{\mathbf{w}}_{t+1}\|^2$. By the definition of $\|\mathcal{G}_\eta(\mathbf{w}_t)\|^2$, $\|\tilde{\mathcal{G}}_\eta(\mathbf{w}_t)\|^2$, we have

$$
\begin{aligned}
\mathbb{E}[\|\mathcal{G}_\eta(\mathbf{w}_t)\|^2] &\leq 2\mathbb{E}[\|\tilde{\mathcal{G}}_\eta(\mathbf{w}_t)\|^2] + \frac{2}{\eta^2}\mathbb{E}[\|\mathbf{w}_{t+1} - \tilde{\mathbf{w}}_{t+1}\|^2] \\
&= 2\mathbb{E}[\|\tilde{\mathcal{G}}_\eta(\mathbf{w}_t)\|^2] + \frac{2}{\eta^2}\mathbb{E}[\|\mathbf{prox}_r^\eta(\mathbf{w}_t - \eta \mathbf{d}_t) - \mathbf{prox}_r^\eta(\mathbf{w}_t - \eta \nabla g(\mathbf{w}_t)^\top \nabla f(g(\mathbf{w}_t)))\|^2] \\
&\leq 2\mathbb{E}[\|\tilde{\mathcal{G}}_\eta(\mathbf{w}_t)\|^2] + \frac{2}{\eta^2}\mathbb{E}[\|\mathbf{w}_t - \eta \mathbf{d}_t - (\mathbf{w}_t - \eta \nabla g(\mathbf{w}_t)^\top \nabla f(g(\mathbf{w}_t)))\|^2] \\
&= 2\mathbb{E}[\|\tilde{\mathcal{G}}_\eta(\mathbf{w}_t)\|^2] + 2\mathbb{E}[\|\nabla g(\mathbf{w}_t)^\top \nabla f(g(\mathbf{w}_t)) - \mathbf{d}_t\|^2],
\end{aligned}
\tag{13}
$$

where the second inequality is due to the non-expansive property of proximal mapping. Similarly, by $\|\mathbf{w}_t - \mathbf{w}_{t+1}\|^2 \leq 2\|\mathbf{w}_t - \tilde{\mathbf{w}}_{t+1}\|^2 + 2\|\mathbf{w}_{t+1} - \tilde{\mathbf{w}}_{t+1}\|^2$, following the same analysis as equation (13), we would have the second inequality in Lemma 5. $\square$

**Lemma 6.** *Let sequence $\{\mathbf{x}_t\}$ be generated by COVER and with $\eta_t \leq \frac{1}{2L}$ for all $t \geq 1$, the following inequality holds*

$$
\mathbb{E}[F(\mathbf{w}_{t+1})] - \mathbb{E}[F(\mathbf{w}_t)] \leq -\frac{\eta_t}{8}\mathbb{E}[\|\mathcal{G}_{\eta_t}(\mathbf{w}_t)\|^2] + \frac{3\eta_t L}{4}\mathbb{E}[\|\varepsilon_{\mathbf{v}_t}\|^2] + \frac{3\eta_t L}{4}\mathbb{E}[\|\varepsilon_{\mathbf{u}_t}\|^2].
\tag{14}
$$

*Proof.* Denote $F(\mathbf{w}_{t+1}) = f(g(\mathbf{w}_{t+1})) + r(\mathbf{w}_{t+1})$. First, show that $f(g(\mathbf{w}))$ is smooth and $\nabla f(\mathbf{w})^\top \nabla f(g(\mathbf{w}))$ has Lipschitz constant with $L_{f(g)} = C_g^2 L_f + C_f L_g$. For any two variables $\mathbf{w}, \mathbf{w}' \in R^d$

$$
\begin{aligned}
&\|\nabla g(\mathbf{w})^\top \nabla f(g(\mathbf{w})) - \nabla g(\mathbf{w}')^\top \nabla f(g(\mathbf{w}'))\| \\
=&\|\nabla g(\mathbf{w})^\top \nabla f(g(\mathbf{w})) - \nabla g(\mathbf{w})^\top \nabla f(g(\mathbf{w}')) + \nabla g(\mathbf{w})^\top \nabla f(g(\mathbf{w}')) - \nabla g(\mathbf{w}')^\top \nabla f(g(\mathbf{w}'))\| \\
\leq&\|\nabla g(\mathbf{w})^\top \nabla f(g(\mathbf{w})) - \nabla g(\mathbf{w})^\top \nabla f(g(\mathbf{w}'))\| + \|\nabla g(\mathbf{w})^\top \nabla f(g(\mathbf{w}')) - \nabla g(\mathbf{w}')^\top \nabla f(g(\mathbf{w}'))\| \\
\leq&\|\nabla g(\mathbf{w})\|\|\nabla f(g(\mathbf{w})) - \nabla f(g(\mathbf{w}'))\| + \|\nabla f(g(\mathbf{w}'))\|\|\nabla g(\mathbf{w}) - \nabla g(\mathbf{w}')\| \\
\leq& C_g L_f \|g(\mathbf{w}) - g(\mathbf{w}')\| + L_g \|\nabla f(g(\mathbf{w}'))\|\|\mathbf{w} - \mathbf{w}'\| \\
\leq& C_g^2 L_f \|\mathbf{w} - \mathbf{w}'\| + L_g C_f \|\mathbf{w} - \mathbf{w}'\| \leq L\|\mathbf{w} - \mathbf{w}'\|.
\end{aligned}
\tag{15}
$$

Then by above equation (15), we have

$$
\begin{aligned}
&f(g(\mathbf{w}_{t+1})) + r(\mathbf{w}_{t+1}) \\
\leq& f(g(\mathbf{w}_t)) + \langle \nabla g(\mathbf{w}_t)^\top \nabla f(g(\mathbf{w}_t)), \mathbf{w}_{t+1} - \mathbf{w}_t \rangle + \frac{L}{2}\|\mathbf{w}_{t+1} - \mathbf{w}_t\|^2 + r(\mathbf{w}_{t+1}) \\
\leq& f(g(\mathbf{w}_t)) + \langle \mathbf{d}_t, \mathbf{w}_{t+1} - \mathbf{w}_t \rangle + r(\mathbf{w}_{t+1}) \\
&+ \langle \nabla g(\mathbf{w}_t)^\top \nabla f(g(\mathbf{w}_t)) - \mathbf{d}_t, \mathbf{w}_{t+1} - \mathbf{w}_t \rangle + \frac{L}{2}\|\mathbf{w}_{t+1} - \mathbf{w}_t\|^2 \\
\overset{(a)}{\leq}& f(g(\mathbf{w}_t)) + r(\mathbf{w}_t) - \frac{1}{2\eta_t}\|\mathbf{w}_{t+1} - \mathbf{w}_t\|^2 + \frac{\eta}{2}\|\nabla g(\mathbf{w}_t)^\top \nabla f(g(\mathbf{w}_t)) - \mathbf{d}_t\|^2 + \frac{L}{2}\|\mathbf{w}_{t+1} - \mathbf{w}_t\|^2 \\
=& F(\mathbf{w}_t) + \frac{\eta_t}{2}\|\nabla g(\mathbf{w}_t)^\top \nabla f(g(\mathbf{w}_t)) - \mathbf{d}_t\|^2 - (\frac{\eta_t}{2} - \frac{L\eta_t^2}{2})\|\tilde{\mathcal{G}}(\mathbf{w}_t)\|^2,
\end{aligned}
\tag{16}
$$

where the proof of $(a)$ will be shown shortly after we derive the claimed result of this lemma. By the setting $\eta_t \leq \frac{1}{2L}$, taking expectation on both sides and in combination with Lemma 5, we have

$$
\mathbb{E}[F(\mathbf{w}_{t+1}) - F(\mathbf{w}_t)] \leq -\frac{\eta_t}{8}\mathbb{E}[\|\mathcal{G}_{\eta_t}(\mathbf{w}_t)\|^2] + \frac{3\eta_t}{4}\|\nabla g(\mathbf{w}_t)^\top \nabla f(g(\mathbf{w}_t)) - \mathbf{d}_t\|^2.
\tag{17}
$$

Then applying the results of Lemma 4, we have the results.

Proof of $(a)$: By the definition of $\mathbf{w}_{t+1} = \mathbf{prox}_r^{\eta_t}(\mathbf{w}_t - \eta_t \mathbf{d}_t) = \arg\min_{\mathbf{w}}\{\frac{1}{2}\|\mathbf{w} - (\mathbf{w}_t - \eta_t \mathbf{d}_t)\|^2 + \eta_t r(\mathbf{w})\} = \arg\min_{\mathbf{w}}\{\frac{1}{2\eta_t}\|\mathbf{w} - (\mathbf{w}_t - \eta_t \mathbf{d}_t)\|^2 + r(\mathbf{w})\}$. Then by the $\frac{1}{\eta_t}$ strongly convexity of the

quadratic function:

$$\frac{1}{2\eta_t}\|\mathbf{w}_{t+1} - (\mathbf{w}_t - \eta_t\mathbf{d}_t)\|^2 + r(\mathbf{w}_{t+1}) \le \frac{1}{2\eta_t}\|\mathbf{w}_t - (\mathbf{w}_t - \eta_t\mathbf{d}_t)\|^2 + r(\mathbf{w}_t) - \frac{1}{2\eta_t}\|\mathbf{w}_{t+1} - \mathbf{w}_t\|^2$$

Then it follows that

$$\langle \mathbf{d}_t, \mathbf{w}_{t+1} - \mathbf{w}_t \rangle + r(\mathbf{w}_{t+1}) \le r(\mathbf{w}_t) - \frac{1}{\eta_t}\|\mathbf{w}_{t+1} - \mathbf{w}_t\|^2.$$

Further by Young's Inequality:

$$\langle \nabla g(\mathbf{w}_t)^\top \nabla f(g(\mathbf{w}_t)) - \mathbf{d}_t, \mathbf{w}_{t+1} - \mathbf{w}_t \rangle \le \frac{\eta_t}{2}\|\nabla g(\mathbf{w}_t)^\top \nabla f(g(\mathbf{w}_t)) - \mathbf{d}_t\|^2 + \frac{1}{2\eta_t}\|\mathbf{w}_{t+1} - \mathbf{w}_t\|^2.$$

$\square$

To prove the convergence of proximal gradient $\|\mathcal{G}_{\eta_t}(\mathbf{w}_t)\|^2$, we need to construct telescoping sum that depending on the Lemma 6. As a result, we need to bound the variance on the R.H.S of Lemma 6, *i.e.*, $\mathbb{E}[\|\varepsilon_{\mathbf{u}_t}\|^2]$, $\mathbb{E}[\|\varepsilon_{\mathbf{v}_t}\|^2]$ with the following lemmas.

**Lemma 7.** *With notations in COVER, we have*

$$\begin{aligned}
\frac{\mathbb{E}[\|\varepsilon_{\mathbf{u}_{t+1}}\|^2]}{\eta_t} &\le \mathbb{E}\Big[2\eta_t^3 c^2\sigma^2 + \frac{(1-a_t)^2(1+4\eta_t^2 L^2)\|\varepsilon_{\mathbf{u}_t}\|^2}{\eta_t} \\
&\quad + \frac{4\eta_t^2(1-a_t)^2 L^2\|\varepsilon_{\mathbf{v}_t}\|^2}{\eta_t} + 4\eta_t(1-a_t)^2 L\|\mathcal{G}_{\eta_t}(\mathbf{w}_t)\|^2\Big] \\
\frac{\mathbb{E}[\|\varepsilon_{\mathbf{v}_{t+1}}\|^2]}{\eta_t} &\le \mathbb{E}\Big[2\eta_t^3 c^2\sigma^2 + \frac{(1-a_t)^2(1+4\eta_t^2 L^2)\|\varepsilon_{\mathbf{v}_t}\|^2}{\eta_t} \\
&\quad + \frac{4\eta_t^2(1-a_t)^2 L^2\|\varepsilon_{\mathbf{u}_t}\|^2}{\eta_t} + 4\eta_t(1-a_t)^2 L\|\mathcal{G}_{\eta_t}(\mathbf{w}_t)\|^2\Big].
\end{aligned} \tag{18}$$

*Proof.*

$$\begin{aligned}
\mathbb{E}[\frac{\|\varepsilon_{\mathbf{u}_{t+1}}\|^2}{\eta_t}] &= \mathbb{E}[\frac{\|\mathbf{u}_{t+1} - g(\mathbf{w}_{t+1})\|^2}{\eta_t}] \\
&= \mathbb{E}\Big[\frac{\|g_{\mathbf{z}_{t+1}}(\mathbf{w}_{t+1}) + (1-a_t)(\mathbf{u}_t - g_{\mathbf{z}_{t+1}}(\mathbf{w}_t)) - g(\mathbf{w}_{t+1})\|^2}{\eta_t}\Big] \\
&= \mathbb{E}\Big[\frac{\|a_t(g_{\mathbf{z}_{t+1}}(\mathbf{w}_{t+1}) - g(\mathbf{w}_{t+1})) + (1-a_t)(\mathbf{u}_t - g(\mathbf{w}_t))}{\eta_t} \\
&\quad + \frac{(1-a_t)(g_{\mathbf{z}_{t+1}}(\mathbf{w}_{t+1}) - g_{\mathbf{z}_{t+1}}(\mathbf{w}_t) - (g(\mathbf{w}_{t+1}) - g(\mathbf{w}_t)))\|^2}{\eta_t}\Big] \\
&= \mathbb{E}\Big[\frac{(1-a_t)^2\|\varepsilon_{\mathbf{u}_t}\|^2}{\eta_t} \\
&\quad + \frac{\|a_t(g_{\mathbf{z}_{t+1}}(\mathbf{w}_{t+1}) - g(\mathbf{w}_{t+1})) + (1-a_t)(g_{\mathbf{z}_{t+1}}(\mathbf{w}_{t+1}) - g_{\mathbf{z}_{t+1}}(\mathbf{w}_t) - (g(\mathbf{w}_{t+1}) - g(\mathbf{w}_t)))\|^2}{\eta_t}\Big] \\
&\le \mathbb{E}\Big[\frac{2a_t^2\|g_{\mathbf{z}_{t+1}}(\mathbf{w}_{t+1}) - g(\mathbf{w}_{t+1})\|^2}{\eta_t} + \frac{(1-a_t)^2\|\varepsilon_{\mathbf{u}_t}\|^2}{\eta_t} \\
&\quad + \frac{2(1-a_t)^2\|g_{\mathbf{z}_{t+1}}(\mathbf{w}_{t+1}) - g_{\mathbf{z}_{t+1}}(\mathbf{w}_t) - (g(\mathbf{w}_{t+1}) - g(\mathbf{w}_t))\|^2}{\eta_t}\Big] \\
&\le \mathbb{E}\Big[\frac{2a_t^2\sigma^2}{\eta_t} + \frac{(1-a_t)^2\|\varepsilon_{\mathbf{u}_t}\|^2}{\eta_t} + \frac{2(1-a_t)^2 L\|\mathbf{w}_{t+1} - \mathbf{w}_t\|^2}{\eta_t}\Big] \\
&= \mathbb{E}\Big[2c^2\eta_t^3\sigma^2 + \frac{(1-a_t)^2\|\varepsilon_{\mathbf{u}_t}\|^2}{\eta_t} + \frac{2(1-a_t)^2 L\eta_t^2\|\tilde{\mathcal{G}}_{\eta_t}(\mathbf{w}_t)\|^2}{\eta_t}\Big] \\
&\le \mathbb{E}\Big[2c^2\eta_t^3\sigma^2 + \frac{(1-a_t)^2\|\varepsilon_{\mathbf{u}_t}\|^2}{\eta_t} + \frac{2(1-a_t)^2 L\eta_t^2}{\eta_t}\Big(2\|\mathcal{G}_{\eta_t}(\mathbf{w}_t)\|^2 + 2L(\|\varepsilon_{\mathbf{u}_t}\|^2 + \|\varepsilon_{\mathbf{v}_t}\|^2)\Big)\Big] \\
&= \mathbb{E}\Big[2\eta_t^3 c^2\sigma^2 + \frac{(1-a_t)^2(1+4\eta_t^2 L^2)\|\varepsilon_{\mathbf{u}_t}\|^2}{\eta_t} + \frac{4\eta_t^2(1-a_t)^2 L(L\|\varepsilon_{\mathbf{v}_t}\|^2 + \|G_{\eta_t}(\mathbf{w}_t)\|^2)}{\eta_t}\Big],
\end{aligned} \tag{19}$$

where the fourth equality is due to $E_t[g_{\mathbf{z}_{t+1}}(\mathbf{w}_{t+1}) - g(\mathbf{w}_{t+1})] = 0$ and $E_t[g_{\mathbf{z}_{t+1}}(\mathbf{w}_t) - g(\mathbf{w}_t)] = 0$ with $E_t$ denoting an expectation conditioned on events until $t$-iteration; and the first inequality holds because $\|a + b\|^2 \leq 2a^2 + 2b^2$. Applying the same analysis, we are able to have the bound of $\mathbb{E}[\frac{\|\varepsilon_{\mathbf{v}_{t+1}}\|^2}{\eta_t}] = \mathbb{E}[\frac{\|\mathbf{v}_{t+1} - \nabla g(\mathbf{w}_{t+1})\|^2}{\eta_t}]$ in the lemma. $\qquad\square$

### 9.1 Proof of Theorem 1

*Proof.* After deriving Lemma 6 and 7 we are ready to prove Theorem 1. We construct Lyapunov function $\Gamma_t = F(\mathbf{w}_t) + \frac{1}{c_0 \eta_{t-1}}[\|\varepsilon_{\mathbf{v}_t}\|^2 + \|\varepsilon_{\mathbf{u}_t}\|^2]$, where $c_0$ is a constant and can be derived in the following proof. According to equation (14)

$$
\begin{aligned}
\mathbb{E}[\Gamma_{t+1} - \Gamma_t] &\leq \mathbb{E}[-\frac{\eta_t}{8}\mathbb{E}[\|\mathcal{G}_{\eta_t}(\mathbf{w}_t)\|^2] + \frac{3\eta_t L}{4}\mathbb{E}[\|\varepsilon_{\mathbf{v}_t}\|^2 + \|\varepsilon_{\mathbf{u}_t}\|^2] \\
&\quad + \frac{1}{c_0 \eta_t}\mathbb{E}[\|\varepsilon_{\mathbf{v}_{t+1}}\|^2 + \|\varepsilon_{\mathbf{u}_{t+1}}\|^2] - \frac{1}{c_0 \eta_{t-1}}\mathbb{E}[\|\varepsilon_{\mathbf{v}_t}\|^2 + \|\varepsilon_{\mathbf{u}_t}\|^2].
\end{aligned}
\tag{20}
$$

Then by telescoping sum from $1, \cdots, T$, and rearranging terms we have

$$
\begin{aligned}
\sum_{t=1}^T \frac{\eta_t}{8}\mathbb{E}[\|\mathcal{G}_{\eta_t}(\mathbf{w}_t)\|^2] &\leq \mathbb{E}[\Gamma_1 - \Gamma_{T+1}] + \underbrace{\sum_{t=1}^T \frac{3\eta_t L}{4}\mathbb{E}[\|\varepsilon_{\mathbf{v}_t}\|^2 + \|\varepsilon_{\mathbf{u}_t}\|^2]}_{\text{ⓐ}} \\
&\quad + \underbrace{\sum_{t=1}^T \frac{1}{c_0 \eta_t}\mathbb{E}[\|\varepsilon_{\mathbf{v}_{t+1}}\|^2 + \|\varepsilon_{\mathbf{u}_{t+1}}\|^2] - \frac{1}{c_0 \eta_{t-1}}\mathbb{E}[[\|\varepsilon_{\mathbf{v}_t}\|^2 + \|\varepsilon_{\mathbf{u}_t}\|^2]}_{\text{ⓑ}}.
\end{aligned}
\tag{21}
$$

We want ⓑ $\leq 0$ such that it can be used to cancel the increasing cumulative variance of term ⓐ.

Next we will upper bound ⓑ up to a negative level:

$$
\begin{aligned}
&\frac{1}{c_0 \eta_t}\mathbb{E}[\|\varepsilon_{\mathbf{v}_{t+1}}\|^2 + \|\varepsilon_{\mathbf{u}_{t+1}}\|^2] - \frac{1}{c_0 \eta_{t-1}}\mathbb{E}[[\|\varepsilon_{\mathbf{v}_t}\|^2 + \|\varepsilon_{\mathbf{u}_t}\|^2] \\
&\overset{Lemma\ 7}{\leq} \frac{1}{c_0}\mathbb{E}[4\eta_t^3 c^2 \sigma^2 + (\frac{(1-a_t)^2(1+8\eta_t^2 L^2)}{\eta_t} - \frac{1}{\eta_{t-1}})[\|\varepsilon_{\mathbf{v}_t}\|^2 + \|\varepsilon_{\mathbf{u}_t}\|^2] \\
&\quad + 8\eta_t(1 - a_{t+1})^2 L\|G_{\eta_t}(\mathbf{w}_t)\|^2] \\
&\leq \frac{1}{c_0}\mathbb{E}[4\underbrace{\eta_t^3 c^2 \sigma^2}_{A_t} + \underbrace{(\frac{(1-a_t)(1+8\eta_t^2 L^2)}{\eta_t} - \frac{1}{\eta_{t-1}})[\|\varepsilon_{\mathbf{v}_t}\|^2 + \|\varepsilon_{\mathbf{u}_t}\|^2]}_{B_t} \\
&\quad + \underbrace{8\eta_t L\|G_{\eta_t}(\mathbf{w}_t)\|^2}_{C_t}].
\end{aligned}
\tag{22}
$$

Next we upper bound $B_t$

$$
B_t \leq (\eta_t^{-1} - \eta_{t-1}^{-1} + \eta_t^{-1}(8\eta_t^2 L^2 - a_t))[\|\varepsilon_{\mathbf{u}_t}\|^2 + \|\varepsilon_{\mathbf{v}_t}\|^2] = (\eta_t^{-1} - \eta_{t-1}^{-1} + \eta_t(8L^2 - c))[\|\varepsilon_{\mathbf{u}_t}\|^2 + \|\varepsilon_{\mathbf{v}_t}\|^2].
\tag{23}
$$

For $\frac{1}{\eta_t} - \frac{1}{\eta_{t-1}}$, by applying $(x+y)^{1/3} - x^{1/3} \leq yx^{-2/3}/3$ and manipulating constant terms, we have

$$
\begin{aligned}
\frac{1}{\eta_t} - \frac{1}{\eta_{t-1}} &= \frac{1}{k}(w + t\sigma^2)^{1/3} - \frac{1}{k}(w + (t-1)\sigma^2)^{1/3} \leq \frac{\sigma^2}{3k(w + (t-1)\sigma^2)^{2/3}} \\
&= \frac{\sigma^2}{3k(w - \sigma^2 + t\sigma^2)^{2/3}} \leq \frac{\sigma^2}{3k(w/2 + t\sigma^2)^{2/3}} \\
&\leq \frac{2^{2/3}\sigma^2}{3k(w + t\sigma^2)^{2/3}} = \frac{2^{2/3}\sigma^2}{3k^3}\eta_t^2 \leq \frac{2^{2/3}}{12Lk^3}\eta_t \leq \frac{\sigma^2}{7Lk^3}\eta_t.
\end{aligned}
\tag{24}
$$

where $w \geq (16Lk)^3$ to have $\eta_t \leq \frac{1}{16L}$. Then by setting $c = 104L^2 + \frac{\sigma^2}{7Lk^3}$,

$$\eta_t(8L^2 - c) \leq -96L^2\eta_t - \sigma^2\eta_t/(7Lk^3).$$

Then we obtain

$$B_t \leq -96L^2\eta_t[\|\varepsilon_{\mathbf{u}_t}\|^2 + \|\varepsilon_{\mathbf{v}_t}\|^2]. \tag{25}$$

Then plugging equation (25) into equation (22) and set $c_0 = 128L$,

$$\frac{1}{128\eta_t L}\mathbb{E}[\|\varepsilon_{\mathbf{v}_{t+1}}\|^2 + \|\varepsilon_{\mathbf{u}_{t+1}}\|^2] - \frac{1}{128\eta_{t-1}L}\mathbb{E}[[\|\varepsilon_{\mathbf{v}_t}\|^2 + \|\varepsilon_{\mathbf{u}_t}\|^2]$$
$$\leq \frac{\eta_t^3 c^2\sigma^2}{32L} - \frac{3L\eta_t}{4}[\|\varepsilon_{\mathbf{u}_t}\|^2 + \|\varepsilon_{\mathbf{v}_t}\|^2] + \frac{\eta_t}{16}\mathbb{E}[\|\mathcal{G}_{\eta_t}(\mathbf{w}_t)\|^2]. \tag{26}$$

Substituting equation (26) into equation (21), Dividing $\eta_t^3$ on both sides of equation (21) and substituting (26). We get

$$\frac{\eta_t}{8}\mathbb{E}[\|\mathcal{G}_{\eta_t}(\mathbf{w}_t)\|^2 \leq \mathbb{E}[\Gamma_t - \Gamma_{t+1}] + \frac{3L\eta_t}{4}\mathbb{E}[\|\varepsilon_{\mathbf{v}_t}\|^2 + \|\varepsilon_{\mathbf{u}_t}\|^2]$$
$$+ \frac{\eta_t^3 c^2\sigma^2}{32L} - \frac{3L\eta_t}{4}[\|\varepsilon_{\mathbf{u}_t}\|^2 + \|\varepsilon_{\mathbf{v}_t}\|^2] + \frac{\eta_t}{16}\mathbb{E}[\|\mathcal{G}_{\eta_t}(\mathbf{w}_t)\|^2]$$
$$\leq \mathbb{E}[\Gamma_t - \Gamma_{t+1}] + \frac{\eta_t^3 c^2\sigma^2}{32L} + \frac{\eta_t}{16}\mathbb{E}[\|\mathcal{G}_{\eta_t}(\mathbf{w}_t)\|^2], \tag{27}$$
$$\sum_{t=1}^T \frac{\eta_t}{16}\mathbb{E}[\|\mathcal{G}_{\eta_t}(\mathbf{w}_t)\|^2] \leq \mathbb{E}[\Gamma_1 - \Gamma_{T+1}] + \sum_{t=1}^T \frac{\eta_t^3 c^2\sigma^2}{32L}.$$

In addition

$$\sum_{t=1}^T \frac{\eta_t^3 c^2\sigma^2}{32L} = \frac{c^2\sigma^2}{32L}\sum_{t=1}^T \frac{k^3}{w + t\sigma^2} \leq \frac{c^2\sigma^2}{32L}\sum_{t=1}^T \frac{k^3}{2\sigma^2 + t\sigma^2} \leq \frac{c^2\sigma^2 k^3}{32L}\ln(T+2), \tag{28}$$

where the first inequality is due to the assumption $w \geq 2\sigma^2$ and the second inequality applies $\sum_{t=1}^T \frac{1}{t+2} \leq \ln(T+2)$.

Then

$$\sum_{t=1}^T \frac{\eta_t}{16}\mathbb{E}[\|\mathcal{G}_{\eta_t}(\mathbf{w}_t)\|^2] \leq \mathbb{E}[\Gamma_1 - \Gamma_{T+1}] + \frac{c^2\sigma^2 k^3}{32L}\ln(T+2),$$
$$\sum_{t=1}^T \eta_t\mathbb{E}[\|\mathcal{G}_{\eta_t}(\mathbf{w}_t)\|^2] \leq 16\mathbb{E}[\Gamma_1 - \Gamma_{T+1}] + \frac{c^2\sigma^2 k^3}{2L}\ln(T+2)$$
$$\leq 16\mathbb{E}[F(\mathbf{w}_1) - F_*] + \frac{16}{c_0\eta_0}\mathbb{E}[\|\varepsilon_{\mathbf{u}_1}\|^2 + \|\varepsilon_{\mathbf{v}_1}\|^2] + \frac{c^2\sigma^2 k^3}{2L}\ln(T+2). \tag{29}$$

Since $\eta_t$ is decreasing, we get

$$\frac{1}{T}\sum_{t=1}^T \mathbb{E}[\|\mathcal{G}_{\eta_t}(\mathbf{w}_t)\|^2] \leq \frac{16(F(\mathbf{w}_1) - F_*)}{\eta_T T} + \frac{16\mathbb{E}[\|\varepsilon_{\mathbf{u}_1}\|^2 + \|\varepsilon_{\mathbf{v}_1}\|^2]}{c_0\eta_0\eta_T T} + \frac{c^2 k^3}{2L}\frac{\ln(T+2)}{T\eta_T}$$
$$\leq O\left(\frac{16(F(\mathbf{w}_1) - F_*)}{T^{2/3}} + \frac{32\sigma^2}{c_0\eta_0 T^{2/3}} + \frac{c^2 k^3}{2L}\frac{\ln(T+2)}{T^{2/3}}\right) \tag{30}$$
$$\leq O\left(\frac{\ln(T+2)}{T^{2/3}}\right),$$

where $O$ suppresses constant scalars.

$\square$

To prove the main Theorem 2, we introduce a new intermediate Theorem 3 for COVER. Compared with Theorem 1, Theorem 3 is developed for a specific scenario of COVER when it has been used for the inner stage of RECOVER, in which a constant step size is used in each stage.

**Theorem 3.** *At $k$-th stage, under the Assumption 1 and 2, let $c \geq 104L^2$ and the step size $\eta_k$, after $T_k$ iterations, the output of RECOVER satisfies,*

$$\mathbb{E}[\|\mathcal{G}_{\eta_k}(\mathbf{w}_t)\|^2] \leq \frac{16(F(\mathbf{w}_{k-1}) - F_*)}{\eta_k T_k} + \frac{c^2\sigma^2\eta_k^2}{2L} + \frac{\mathbb{E}[\|\mathbf{u}_{k-1} - g(\mathbf{w}_{k-1})\|^2 + \|\mathbf{v}_{k-1} - \nabla g(\mathbf{w}_{k-1})\|^2]}{8\eta_k^2 L T_k}$$

(31)

*where $\mathbf{w}_k$ is uniformly sampled from $\{\mathbf{w}_t\}_{t=1}^{T_k}$ at $k$-th stage.*

## 9.2 Proof of Theorem 3

*Proof of Theorem 3.* We derive the theoretical analysis for the $k$-th stage based on Lemma 6 and 7. We construct Lyapunov function $\Gamma_t = F(\mathbf{w}_t) + \frac{1}{c_0\eta}[\|\varepsilon_{\mathbf{v}_t}\|^2 + \|\varepsilon_{\mathbf{u}_t}\|^2]$, where $c_0$ is a constant and can be derived in the following proof. According to equation (14)

$$\mathbb{E}[\Gamma_{t+1} - \Gamma_t] \leq \mathbb{E}[-\frac{\eta_k}{8}\mathbb{E}[\|\mathcal{G}_{\eta_k}(\mathbf{w}_t)\|^2] + \frac{3\eta_k L}{4}\mathbb{E}[\|\varepsilon_{\mathbf{v}_t}\|^2 + \|\varepsilon_{\mathbf{u}_t}\|^2]$$

$$+ \frac{1}{c_0\eta_k}\mathbb{E}[\|\varepsilon_{\mathbf{v}_{t+1}}\|^2 + \|\varepsilon_{\mathbf{u}_{t+1}}\|^2] - \frac{1}{c_0\eta_k}\mathbb{E}[\|\varepsilon_{\mathbf{v}_t}\|^2 + \|\varepsilon_{\mathbf{u}_t}\|^2].$$

(32)

Then by telescoping sum and rearranging terms we have

$$\sum_{t=1}^{T_k} \frac{\eta_k}{8}\mathbb{E}[\|\mathcal{G}_{\eta_k}(\mathbf{w}_t)\|^2] \leq \mathbb{E}[\Gamma_1 - \Gamma_{T_{k+1}}] + \underbrace{\sum_{t=1}^{T_k} \frac{3\eta_k L}{4}\mathbb{E}[\|\varepsilon_{\mathbf{v}_t}\|^2 + \|\varepsilon_{\mathbf{u}_t}\|^2]}_{\textcircled{a}}$$

$$+ \underbrace{\sum_{t=1}^{T_k} \frac{1}{c_0\eta_k}\mathbb{E}[\|\varepsilon_{\mathbf{v}_{t+1}}\|^2 + \|\varepsilon_{\mathbf{u}_{t+1}}\|^2] - \frac{1}{c_0\eta_k}\mathbb{E}[\|\varepsilon_{\mathbf{v}_t}\|^2 + \|\varepsilon_{\mathbf{u}_t}\|^2]}_{\textcircled{b}}.$$

(33)

As a result, we want $\textcircled{b} \leq 0$ such that it can be used to cancel the increasing cumulative variance of term $\textcircled{a}$.

Next we will upper bound $\textcircled{b}$ up to a negative level by making use of Lemma 7 with $a_t$ to be fixed at $k$-th stage as $a_k = c\eta_k^2$.

Applying Lemma 7,

$$\frac{1}{c_0\eta_k}\mathbb{E}[\|\varepsilon_{\mathbf{v}_{t+1}}\|^2 + \|\varepsilon_{\mathbf{u}_{t+1}}\|^2] - \frac{1}{c_0\eta_k}\mathbb{E}[[\|\varepsilon_{\mathbf{v}_t}\|^2 + \|\varepsilon_{\mathbf{u}_t}\|^2]$$

$$\leq \frac{1}{c_0}\mathbb{E}\left[4\eta_k^3 c^2\sigma^2 + \left(\frac{(1-a)^2(1+8\eta_k^2 L^2)}{\eta_k} - \frac{1}{\eta_k}\right)[\|\varepsilon_{\mathbf{v}_t}\|^2 + \|\varepsilon_{\mathbf{u}_t}\|^2]\right.$$

$$\left. + 8\eta_k(1-a)^2 L\|G_{\eta_k}(\mathbf{w}_t)\|^2\right]$$

$$\leq \frac{1}{c_0}\mathbb{E}\left[\underbrace{4\eta_k^3 c^2\sigma^2}_{A_t} + \underbrace{\left(\frac{(1-a)(1+8\eta_k^2 L^2)}{\eta_k} - \frac{1}{\eta_k}\right)[\|\varepsilon_{\mathbf{v}_t}\|^2 + \|\varepsilon_{\mathbf{u}_t}\|^2]}_{B_t}\right.$$

$$\left. + \underbrace{8\eta_k L\|G_{\eta_k}(\mathbf{w}_t)\|^2}_{C_t}\right].$$

(34)

For $B_t$, by set $c = 104L^2$, we have

$$B_t \leq (\eta_k^{-1} - \eta_k^{-1} + \eta_k^{-1}(8\eta_k^2 L^2 - a)[\|\varepsilon_{\mathbf{u}_t}\|^2 + \|\varepsilon_{\mathbf{v}_t}\|^2]$$

$$= (\eta_k^{-1} - \eta_k^{-1} + \eta_k(8L^2 - c))[\|\varepsilon_{\mathbf{u}_t}\|^2 + \|\varepsilon_{\mathbf{v}_t}\|^2] \leq -96L^2\eta_k[\|\varepsilon_{\mathbf{u}_t}\|^2 + \|\varepsilon_{\mathbf{v}_t}\|^2].$$

(35)

To satisfies $c\eta_k^2 \le 1$, we should have $\eta_k \le \frac{1}{16L}$. Then by setting $c_0 = 128L$, we have

$$
\sum_{t=1}^{T_k-1} \left[ \frac{1}{128\eta_k L} \mathbb{E}[\|\varepsilon_{\mathbf{v}_{t+1}}\|^2 + \|\varepsilon_{\mathbf{u}_{t+1}}\|^2] - \frac{1}{128\eta_k L} \mathbb{E}[\|\varepsilon_{\mathbf{v}_t}\|^2 + \|\varepsilon_{\mathbf{u}_t}\|^2] \right]
$$

$$
\le \frac{\eta_k^3 c^2 \sigma^2 T_k}{32L} - \sum_{t=1}^{T_k} \frac{3L\eta_k}{4} \mathbb{E}[\|\varepsilon_{\mathbf{u}_t}\|^2 + \|\varepsilon_{\mathbf{v}_t}\|^2] + \sum_{t=1}^{T_k} \frac{\eta_k}{16} \mathbb{E}[\|\mathcal{G}_{\eta_k}(\mathbf{w}_t)\|^2]. \tag{36}
$$

Plugging it into equation (33), we get

$$
\mathbb{E}\left[ \frac{\eta_k}{8} \sum_{t=1}^{T_k} \|\mathcal{G}_{\eta_k}(\mathbf{w}_t)\|^2 \right] \le \mathbb{E}[\Gamma_1 - \Gamma_{T_k+1}] + \mathbb{E}\left[ \frac{c^2\sigma^2}{32L} \eta_k^3 T_k + \frac{\eta_k}{16} \sum_{t=1}^{T_k} \|\mathcal{G}_{\eta_k}(\mathbf{w}_t)\|^2 \right]. \tag{37}
$$

Then we have

$$
\mathbb{E}\left[ \frac{\eta_k}{16} \sum_{t=1}^{T_k} \|\mathcal{G}_{\eta_k}(\mathbf{w}_t)\|^2 \right] \le \frac{c^2\sigma^2}{32L} \eta_k^3 T_k + \mathbb{E}[\Gamma_1 - \Gamma_{T_k+1}]
$$

$$
\le \mathbb{E}[F(\mathbf{w}_1) - F_*] + \frac{c^2\sigma^2\eta_k^3 T_k}{32L} + \frac{\mathbb{E}[\|\varepsilon_{\mathbf{v}_1}\|^2 + \|\varepsilon_{\mathbf{u}_1}\|^2]}{128\eta_k L} \tag{38}
$$

$$
\iff \mathbb{E}[\|\mathcal{G}_{\eta_k}(\mathbf{w}_k)\|^2] \le \frac{16\mathbb{E}[F(\mathbf{w}_1) - F_*]}{\eta_k T_k} + \frac{c^2\sigma^2}{2L}\eta_k^2 + \frac{\mathbb{E}[\|\varepsilon_{\mathbf{v}_1}\|^2 + \|\varepsilon_{\mathbf{u}_1}\|^2]}{8\eta_k^2 L T_k}.
$$

where $\mathbf{w}_k$ is uniformly sampled from $\{\mathbf{w}_1, \cdots, \mathbf{w}_T\}$. $\qquad\square$

## 10 Proof of Section 5

### 10.1 Proof of Lemma 1

*Proof.* This proof follows Lemma A.3 of [50]. Note that

$$
F_{dro}(\mathbf{w}) = \lambda \log \left( \frac{1}{n} \sum_{i=1}^n \exp\left( \frac{\ell(\mathbf{w}; \mathbf{z}_i)}{\lambda} \right) \right)
$$

$$
= \max_{\mathbf{p} \in \Delta_n} (F_{\mathbf{p}}(\mathbf{w}) - h(\mathbf{p}, \mathbf{1}/n)). \tag{39}
$$

Denote $\psi(\mathbf{w}, \mathbf{p}) = F_{\mathbf{p}}(\mathbf{w}) - h(\mathbf{p}, \mathbf{1}/n)$ and $p^*(\mathbf{w}) = \arg\max_{\mathbf{p} \in \Delta_n} \psi(\mathbf{w}, \mathbf{p})$.

Thus, we have $F_{dro}(\mathbf{w}) = \max_{\mathbf{p} \in \Delta_n} \psi(\mathbf{w}, \mathbf{p}) = \psi(\mathbf{w}, p^*(\mathbf{w}))$. By Lemma 4.3 of [26], we know $\nabla F_{dro}(\mathbf{w}) = \nabla_{\mathbf{w}} \psi(\mathbf{w}, p_*(\mathbf{w})) = \nabla_{\mathbf{w}} F_{p^*(\mathbf{w})}(\mathbf{w})$.

Since $F_{\mathbf{p}}(\mathbf{w})$ satisfies a $\mu$-PL condition for any $\mathbf{p} \in \Delta_n$, we have

$$
\|\nabla F_{dro}(\mathbf{w})\|^2 = \|\nabla F_{p^*(\mathbf{w})}(\mathbf{w})\|^2
$$

$$
\ge 2\mu \left( F_{p^*(\mathbf{w})}(\mathbf{w}) - \min_{\mathbf{w}'} F_{p^*(\mathbf{w})}(\mathbf{w}') \right) \tag{40}
$$

$$
= 2\mu \left( \psi(\mathbf{w}, p^*(\mathbf{w})) - \min_{\mathbf{w}'} \psi(\mathbf{w}', p^*(\mathbf{w})) \right).
$$

For any $\mathbf{w}'$,

$$
\psi(\mathbf{w}', p^*(\mathbf{w})) \le \max_{\mathbf{p}'} \psi(\mathbf{w}', \mathbf{p}'). \tag{41}
$$

Therefore,

$$
\min_{\mathbf{w}'} \psi(\mathbf{w}', p^*(\mathbf{w})) \le \min_{\mathbf{w}'} \max_{\mathbf{p}'} \psi(\mathbf{w}', \mathbf{p}'). \tag{42}
$$

Plug this into (40), we get

$$
\|\nabla F_{dro}(\mathbf{w})\|^2 \ge 2\mu \left( \psi(\mathbf{w}, p^*(\mathbf{w})) - \min_{\mathbf{w}'} \max_{\mathbf{p}'} \psi(\mathbf{w}', \mathbf{p}') \right)
$$

$$
= 2\mu(F_{dro}(\mathbf{w}) - \min_{\mathbf{w}'} F_{dro}(\mathbf{w}')), \tag{43}
$$

which means $F_{dro}$ satisfies the $\mu$-PL condition. $\qquad\square$

## 10.2 Proof of Lemma 2

*Proof.* Let us define scaled data $\mathbf{v}_i = \sqrt{p_i}\mathbf{x}_i, 1 \le i \le n$ with $p_i \ge p_0$. Then we have $\|\mathbf{v}_i - \mathbf{v}_j\| \ge \sqrt{p_0}\delta$ since $p_i \ge p_0$.

Taking $\{(\mathbf{v}_1, \sqrt{p_i}y_1), ..., (\mathbf{v}_n, \sqrt{p_i}y_n)\}$ as input to the defined network, then we accordingly denote the output of the first layer of the network as $\hat{h}_{i,0} = \phi(\sqrt{p_i}A\mathbf{x}_i) = \sqrt{p_i}\phi(A\mathbf{x}_i) = \sqrt{p_i}h_{i,0}$, where the the second equality is due to the property of ReLU activation function. By induction, we see that the output of the $l$-th layer is $\hat{h}_{i,l} = \sqrt{p_i}h_{i,l}$. And then the output logit is $\hat{y}_i(\mathbf{v}_i) = \sqrt{p_i}\hat{y}_i$.

As a result, the weighted loss defined on the original data is the average of square loss on the scaled data,

$$F(W, \mathbf{p}) = \frac{1}{n}\sum_{i=1}^{n}\ell(W; \mathbf{v}_i) = \frac{1}{n}\sum_{i=1}^{n}(\sqrt{p_i}\hat{y}_i - \sqrt{p_i}y_i)^2 = \sum_{i=1}^{n}p_i(\hat{y}_i - y_i)^2 \tag{44}$$

Then we plug in Theorem 3, Lemma 7.4 and Lemma 8.7 of [1] with $F(W)$ as the objective function and $\{(\mathbf{v}_1, \sqrt{p_i}y_1), ..., (\mathbf{v}_n, \sqrt{p_i}y_n)\}$ as input data. We obtain that for any fixed $\mathbf{p} \in \Delta, p_i \ge p_0$, with probability $1 - \exp(-\Omega(d_2/\mathrm{poly}(n, \tilde{L}, \delta^{-1})))$, it holds for every $W$ with $\|W - W_0\|^2 \le \frac{1}{\mathrm{poly}(n, \tilde{L}, \delta^{-1})}$,

$$\|\nabla_W F(W, \mathbf{p})\|_F^2 \ge \Omega\left(\frac{\sqrt{p_0}\delta d_2}{d_0 n^2}(F(W, \mathbf{p}) - \min_{W'} F(W', \mathbf{p}))\right), \tag{45}$$

and

$$\|\hat{y}_i - y_i\|^2 \le \mathrm{poly}(d_2, d_0^{-1}, \tilde{L}), \|2(\hat{y}_i - y_i)\nabla_W \hat{y}_i\| \le \mathrm{poly}(d_2, d_0^{-1}, \tilde{L}). \tag{46}$$

To generalize this bound to all $\mathbf{p} \in \Delta, p_i \ge p_0$, we need to introduce $\epsilon$-net. A subset $\mathcal{N} \subset \mathcal{K}$ is called an $\epsilon$-net of $\mathcal{K}$ if for every $\mathbf{w} \in \mathcal{K}$ one can find $\tilde{\mathbf{w}} \in \mathcal{N}$ so that $\|\mathbf{w} - \tilde{\mathbf{w}}\| \le \epsilon$. Let $\mathcal{N}(\mathcal{K}, \epsilon)$ denote the $\epsilon$-net of a set $\mathcal{K}$ with minimal cardinality, which is referred to as the covering number. It can be seen that the set $\mathcal{P} = \{\mathbf{p}|\mathbf{p} \in \Delta, p_i \ge p_0\}$ can be covered by a $n$-dimension unit ball $\mathcal{B}$. Take $\epsilon' = O(\epsilon/\mathrm{poly}(d_2, d_0^{-1}, \tilde{L}))$. According to a standard volume comparison argument [37], we have

$$\log|\mathcal{N}(\mathcal{B}, \epsilon')| \le n\log\frac{3}{\epsilon'}. \tag{47}$$

Since we have $\mathcal{P} \subset \mathcal{B}$, it follows that

$$\log|\mathcal{N}(\mathcal{P}, \epsilon')| \le \log|\mathcal{N}(\mathcal{B}, \frac{\epsilon'}{2})| \le n\log\frac{6}{\epsilon'}, \tag{48}$$

where the first inequality is due to that the covering numbers are (almost) increasing by inclusion [38]. Taking union bound over the $\epsilon'$-net $\mathcal{N}(\mathcal{P}, \epsilon')$, we obtain that with probability $1 - \exp(-\tilde{\Omega}(d_2/\mathrm{poly}(n, \tilde{L}, \delta^{-1})))$, it holds for every $\mathbf{p} \in \mathcal{N}(\mathcal{P}, \epsilon')$ and for every $W$ with $\|W - W_0\|^2 \le \frac{1}{\mathrm{poly}(n, \tilde{L}, \delta^{-1})}$,

$$\|\nabla_W F(W, \mathbf{p})\|_F^2 \ge \Omega\left(\frac{\sqrt{p_0}\delta d_2}{d_0 n^2}(F(W, \mathbf{p}) - \min_{W'} F(W', \mathbf{p}))\right), \tag{49}$$

and

$$\|\hat{y}_i - y_i\|^2 \le \mathrm{poly}(d_2, d_0^{-1}, \tilde{L}), \|2(\hat{y}_i - y_i)\nabla_W \hat{y}_i\| \le \mathrm{poly}(d_2, d_0^{-1}, \tilde{L}). \tag{50}$$

For $\mathbf{p}$ not in $\mathcal{N}(\mathcal{P}, \epsilon')$, let $\hat{\mathbf{p}}$ be a point in $\mathcal{N}(\mathcal{P}, \epsilon')$ such that $\|\hat{\mathbf{p}} - \mathbf{p}\| \le \epsilon'$, we have

$$\begin{aligned}
2\|\nabla_W F(W, \mathbf{p})\|_F^2 + O(\epsilon) &\ge 2\|\nabla_W F(W, \mathbf{p})\|_F^2 + 2\|\nabla_W F(W, \mathbf{p}) - \nabla_W F(W, \hat{\mathbf{p}})\|_F^2 \\
&\ge \|\nabla_W F(W, \hat{\mathbf{p}})\|_F^2 \ge \Omega\left(\frac{\sqrt{p_0}\delta d_2}{d_0 n^2}(F(W, \hat{\mathbf{p}}) - \min_{W'} F(W', \hat{\mathbf{p}}))\right) \\
&\ge \Omega\left(\frac{\sqrt{p_0}\delta d_2}{d_0 n^2}(F(W, \mathbf{p}) - \min_{W'} F(W', \mathbf{p}))\right) - O(\epsilon),
\end{aligned} \tag{51}$$

where the first inequality uses the second part of (50) and $\epsilon' = O(\epsilon/\mathrm{poly}(d_2, d_0^{-1}, \tilde{L}))$, and the last inequality uses the first part of (50).

We also have

$$F_{dro}(W) = \max_{\mathbf{p}\in\Delta, p_i\geq p_0} F(W,\mathbf{p}) - h(\mathbf{p},1/n) = F(W,p^*(W)) - h(p^*(W),1/n),$$
$$\nabla F_{dro}(W) = \nabla_W F(W,p^*(W)),$$

(52)

where the second line uses standard property of min-max problem [26]. Thus (45) implies that, with probability $1 - \exp(-\tilde{\Omega}(d_2/\mathrm{poly}(n,\tilde{L},\delta^{-1})))$, it holds for every $W$ with $\|W - W_0\|^2 \leq \frac{1}{\mathrm{poly}(n,\tilde{L},\delta^{-1})}$,

$$\|\nabla F_{dro}(W)\|_F^2 + O(\epsilon)$$
$$\geq \Omega\left(\frac{\sqrt{p_0}\delta d_2}{d_0 n^2}\left(F(W,p^*(W)) - h(p^*(W),1/n) - \min_{W'}(F(W',p^*(W)) - h(p^*(W),1/n))\right)\right)$$
$$\geq \Omega\left(\frac{\sqrt{p_0}\delta d_2}{d_0 n^2}\left(F(W,p^*(W)) - h(p^*(W),1/n) - \min_{W'}\max_{p'}(F(W',p') - h(p',1/n))\right)\right)$$
$$\geq \Omega\left(\frac{\sqrt{p_0}\delta d_2}{d_0 n^2}(F_{dro}(W) - \min_{W'} F_{dro}(W'))\right),$$

(53)

where the second inequality holds due to the same reason as (41) and (42).

This means that $F_{dro}(W)$ satisfies a $\mu$-PL condition with $\mu \in O\left(\frac{\sqrt{p_0}\delta d_2}{d_0 n^2}\right)$ with an extra addition term of $O(\epsilon)$, which will be omitted later in the paper for simplicity. $\qquad\square$

### 10.3  Reduced Variance (Proof of Lemma 3)

*Proof.* This lemma implies that the variance also decreasing with the increasing of stages. By equation (36) and rearranging terms, the cumulative variance of $k$-th stage satisfies:

$$\mathbb{E}[\sum_{t=1}^{T_k}\frac{3L\eta_k}{4}[\|\varepsilon_{\mathbf{u}_t}\|^2 + \|\varepsilon_{\mathbf{v}_t}\|^2]] \leq \frac{1}{128\eta_k L}\mathbb{E}[\|\varepsilon_{\mathbf{v}_1}\| + \|\varepsilon_{\mathbf{u}_1}\|^2] + \frac{\eta_k^3 c^2\sigma^2 T_k}{32L} + \sum_{t=1}^{T_k}\frac{\eta_k}{16}\mathbb{E}[\|\mathcal{G}_{\eta_k}(\mathbf{w}_t)\|^2]$$
$$\leq \mathbb{E}[F(\mathbf{w}_1) - F_*] + \frac{c^2\sigma^2\eta_k^3 T_k}{4L} + \frac{\mathbb{E}[\|\varepsilon_{\mathbf{v}_1}\|^2 + \|\varepsilon_{\mathbf{u}_1}\|^2]}{64\eta_k L},$$

(54)

where the second inequality uses Theorem 3. Thus we have,

$$\mathbb{E}[\|\varepsilon_{\mathbf{u}_\tau}\|^2 + \|\varepsilon_{\mathbf{v}_\tau}\|^2] \leq \frac{2\mathbb{E}[F(\mathbf{w}_1) - F_*]}{\eta_k T_k L} + \frac{c^2\sigma^2\eta_k^2}{3L^2} + \frac{\mathbb{E}[\|\varepsilon_{\mathbf{v}_1}\|^2 + \|\varepsilon_{\mathbf{u}_1}\|^2]}{48\eta_k^2 L^2 T_k},$$

(55)

where $\tau$ is randomly sampled from $1,\cdots,T_k$.

Without loss of generality, let's assume that $\epsilon_0 = \Delta_F \geq \frac{c^2\sigma^2}{64\mu L^4}$, i.e., $\frac{\sqrt{\mu\epsilon_0}L}{2c\sigma} \geq \frac{1}{16L}$. The case that $\Delta_F < \frac{c^2\sigma^2}{64\mu L^4}$ can be simply covered by our proof. Then, denote $\epsilon_1 = \frac{c^2\sigma^2}{64\mu L^4}$ and $\epsilon_k = \epsilon_1/2^{k-1}$, $c = 104L^2$.

Let's consider the first stage, we have initialization such that $F(\mathbf{w}_1) - F_* = \Delta_F$ and $\mathbb{E}[\|\varepsilon_{\mathbf{v}_1}\|^2 + \|\varepsilon_{\mathbf{u}_1}\|^2] \leq \sigma^2$. Setting $\eta_1 = \frac{1}{16L}$ and $T_1 = O(\max(\frac{\Delta_F}{\sigma^2},1))$. Note that in below the numerical subscripts denote the epoch index $(1,...,K)$. We bound the the error of first stage's output as follows,

$$\mathbb{E}[\|\varepsilon_{\mathbf{u}_1}\|^2 + \|\varepsilon_{\mathbf{v}_1}\|^2] \leq \frac{2\mathbb{E}[F(\mathbf{w}_1) - F_*]}{\eta_1 T_1 L} + \frac{c^2\sigma^2\eta_1^2}{3L^2} + \frac{\mathbb{E}[\|\varepsilon_{\mathbf{v}_1}\|^2 + \|\varepsilon_{\mathbf{u}_1}\|^2]}{48\eta_1^2 L^2 T_1}$$
$$= \frac{2\mathbb{E}[F(\mathbf{w}_1) - F_*]}{\eta_1 T_1 L} + \frac{c^2\sigma^2\eta_1^2}{3L^2} + \frac{\mathbb{E}[\|\varepsilon_{\mathbf{v}_1}\|^2 + \|\varepsilon_{\mathbf{u}_1}\|^2]}{48\eta_1^2 L^2 T_1}$$
$$\leq \frac{2\epsilon_0}{\eta_1 T_1 L} + \frac{c^2\sigma^2\eta_1^2}{3L^2} + \frac{\sigma^2}{24\eta_1^2 L^2 T_1} \leq \frac{c^2\sigma^2}{64L^4} = \mu\epsilon_1.$$

(56)

Starting from the second stage, we will prove by induction. Suppose we are at $k$-th stage. Assuming that $F(\mathbf{w}_{k-1}) - F_* \leq \epsilon_{k-1}$ and $\|\varepsilon_{\mathbf{v}_{k-1}}\|^2 + \|\varepsilon_{\mathbf{u}_{k-1}}\|^2 \leq \mu\epsilon_{k-1}$ after the $(k-1)$-th stage, we will

show that $\mathbb{E}[\|\varepsilon_{\mathbf{u}_k}\|^2 + \|\varepsilon_{\mathbf{v}_k}\|^2] \le \mu\epsilon_k$ by induction. Note that the induction of $F(\mathbf{w}_k) - F(\mathbf{w}_0)$ will be addressed later in Theorem 2.

$$
\begin{aligned}
\mathbb{E}[\|\varepsilon_{\mathbf{u}_k}\|^2 + \|\varepsilon_{\mathbf{v}_k}\|^2] &\le \frac{2\mathbb{E}[F(\mathbf{w}_{k-1}) - F_*]}{\eta_k T_k L} + \frac{c^2\sigma^2\eta_k^2}{3L^2} + \frac{\mathbb{E}[\|\varepsilon_{\mathbf{v}_{k-1}}\|^2 + \|\varepsilon_{\mathbf{u}_{k-1}}\|^2]}{48\eta_k^2 L^2 T_k} \\
&= \frac{2\mathbb{E}[F(\mathbf{w}_{k-1}) - F_*]}{\eta_k T_k L} + \frac{c^2\sigma^2\eta_k^2}{3L^2} + \frac{\mathbb{E}[\|\varepsilon_{\mathbf{v}_{k-1}}\|^2 + \|\varepsilon_{\mathbf{u}_{k-1}}\|^2]}{48\eta_k^2 L^2 T_k} \\
&\le \frac{2\epsilon_{k-1}}{\eta_k T_k L} + \frac{c^2\sigma^2\eta_k^2}{3L^2} + \frac{\mu\epsilon_{k-1}}{\eta_k^2 L^2 T_k} \le \frac{\mu\epsilon_{k-1}}{2} = \mu\epsilon_k,
\end{aligned}
\tag{57}
$$

where the last inequality follows from the setting that $\eta_k = \frac{\sqrt{\mu\epsilon_k}L}{2c\sigma} \le \frac{1}{16L}$, and $T_k = \max\{\frac{96c\sigma}{\mu^{3/2}\sqrt{\epsilon_k}L^2}, \frac{16c^2\sigma^2}{\mu L^4\epsilon_k}\}$, where $c = 104L^2$.

$\square$

## 10.4 Poof of Theorem 2

*Proof.* Without loss of generality, let's assume that $\epsilon_0 = \Delta_F \ge \frac{c^2\sigma^2}{64\mu L^4}$, i.e., $\frac{\sqrt{\mu\epsilon_0}L}{2c\sigma} \ge \frac{1}{16L}$. The case that $\Delta_F < \frac{c^2\sigma^2}{64\mu L^4}$ can be simply covered by our proof. Then, denote $\epsilon_1 = \frac{c^2\sigma^2}{64\mu L^4}$ and $\epsilon_k = \epsilon_1/2^{k-1}$, $c = 104L^2$.

Note that in below the numerical subscripts denote the epoch index $(1, ..., K)$ (different from in proof of Lemma 4 which all are in one stage). Let's consider the first stage, we have initialization such that $F(\mathbf{w}_0) - F_* = \Delta_F$ and $\mathbb{E}[\|\varepsilon_{\mathbf{v}_0}\|^2 + \|\varepsilon_{\mathbf{u}_0}\|^2] \le \sigma^2$. We bound the the error of first stage's output as follows,

$$
\begin{aligned}
\mathbb{E}[F(\mathbf{w}_1) - F_*] &\le \frac{1}{2\mu}\mathbb{E}[\|\mathcal{G}_{\eta_1}(\mathbf{w}_1)\|^2] \\
&\le \frac{8\mathbb{E}[F(\mathbf{w}_0) - F(\mathbf{w}_*)]}{\mu\eta_1 T_1 L} + \frac{c^2\sigma^2}{4\mu L^2}\eta_1^2 + \frac{\mathbb{E}[\|\varepsilon_{\mathbf{v}_0}\|^2 + \|\varepsilon_{\mathbf{u}_0}\|^2]}{16\mu\eta_1^2 L T_1} \\
&\le \frac{8\Delta_F}{\mu\eta_1 T_1 L} + \frac{c^2\sigma^2}{4\mu L^2}\eta_1^2 + \frac{\sigma^2}{8\mu\eta_1^2 L T_1} \le \frac{c^2\sigma^2}{64\mu L^4} = \epsilon_1,
\end{aligned}
\tag{58}
$$

where the first inequality uses PL condition, the second inequality use Theorem 3 and the fourth inequality uses the setting of $\eta_1 = \frac{1}{16L}$ and $T_1 = O(\max(\frac{\Delta_F}{\sigma^2}, 1))$.

Starting from the second stage, we will prove by induction. Denote $\epsilon_1 = \frac{c^2\sigma^2}{64\mu L^4}$ and $\epsilon_k \le \epsilon_1/2^{k-1}$ for $k \ge 2$. Suppose at the beginning of $k$-stage ($k \ge 2$), we have $\mathbb{E}[F(\mathbf{w}_{k-1}) - F_*] \le \epsilon_{k-1}$ and $\mathbb{E}[\|\epsilon_{\mathbf{v}_{k-1}}\|^2 + \|\epsilon_{\mathbf{u}_{k-1}}\|^2] \le \mu\epsilon_{k-1}$. When $k \ge 2$, we have $\eta_k = \frac{\sqrt{\mu\epsilon_k}L}{2c\sigma} \le \frac{1}{16L}$. Then by Lemma 3 and Theorem 3, setting $T_k = \max\{\frac{96c\sigma}{\mu^{3/2}\sqrt{\epsilon_k}L}, \frac{16c^2\sigma^2}{\mu L^2\epsilon_k}\}$, RECOVER satisfies the following equations at the $k$-th stage,

$$
\begin{aligned}
\mathbb{E}[F(\mathbf{w}_k) - F_*] &\le \frac{1}{2\mu}\mathbb{E}[\|\mathcal{G}_\eta(\mathbf{w}_k)\|^2] \\
&\le \frac{8\mathbb{E}[F(\mathbf{w}_{k-1}) - F(\mathbf{w}_*)]}{\mu\eta_k T_k} + \frac{c^2\sigma^2}{4\mu L}\eta_k^2 + \frac{\mathbb{E}[\|\varepsilon_{\mathbf{v}_{k-1}}\|^2 + \|\varepsilon_{\mathbf{u}_{k-1}}\|^2]}{16\mu\eta_k^2 L T_k} \\
&\le \frac{8\mathbb{E}[F(\mathbf{w}_{k-1}) - F_*]}{\mu\eta_k T_k} + \frac{c^2\sigma^2}{4\mu L}\eta_k^2 + \frac{\|\varepsilon_{\mathbf{v}_{k-1}}\|^2 + \|\varepsilon_{\mathbf{u}_{k-1}}\|^2}{16\mu\eta_k^2 L T_k} \\
&\le \frac{8\epsilon_{k-1}}{\mu\eta_k T_k} + \frac{c^2\sigma^2}{4\mu L}\eta_k^2 + \frac{\mu\epsilon_{k-1}}{16\mu\eta_k^2 L T_k} \\
&\le \frac{\epsilon_{k-1}}{2} \le \epsilon_k,
\end{aligned}
\tag{59}
$$

where the forth inequality is implied by the induction hypothesis and the last inequality holds by the setting of $\eta_k$ and $T_k$.

Combing two cases, after $K \leq 1 + \log_2(\epsilon_1/\epsilon) \leq \log_2(\epsilon_0/\epsilon)$ stages, $\mathbb{E}[F(\mathbf{w}_k) - F(\mathbf{w}_*)] \leq \epsilon$.

By setting $c = 104L^2$, following th the proof of Theorem 2, the sample complexity of RECOVER equals to the number of samples in the first stage, i.e., $T_1$ plus the number of samples in later stages, i.e. $\sum_{k=2}^{K} T_k$, which is

$$
T_1 + \sum_{k=2}^{K} T_k = O\left(\frac{\Delta_F}{\sigma^2} + \sum_{k=2}^{K} T_k\right)
$$

$$
= O\left(\sum_{k=2}^{K}\left(\frac{c\sigma}{\mu^{3/2}\sqrt{\epsilon_k}L} + \frac{L^2\sigma^2}{\mu\epsilon_k}\right)\right)
$$

$$
\leq O\left(\frac{c\sigma}{\mu^{3/2}\sqrt{\epsilon}L} + \frac{c^2\sigma^2}{L^2\mu\epsilon}\right) \overset{\mu \geq \epsilon}{\leq} O\left(\frac{1}{\mu\epsilon}\right).
$$

$\square$

## 11    Derivation of the Compositional Formulation

Recall the problem:

$$
\min_{\mathbf{w}\in\mathbb{R}^d} \max_{\mathbf{p}\in\Delta_n} F_{\mathbf{p}}(\mathbf{w}) = \sum_{i=1}^{n} p_i\ell(\mathbf{w};\mathbf{z}_i) - h(\mathbf{p},\mathbf{1}/n) + r(\mathbf{w}),
$$

where $\Delta_n = \{\mathbf{p}\in\mathbb{R}^n : \sum_i p_i = 1, 0 \leq p_i \leq 1\}$. In order to solve the inner maximization, we will fix $\mathbf{w}$ and derive an optimal solution $\mathbf{p}^*(\mathbf{w})$ that depends on $\mathbf{w}$. To this end, we consider the following problem:

$$
\min_{\mathbf{p}\in\Delta_n} - \sum_{i=1}^{n} p_i\ell(\mathbf{w};\mathbf{z}_i) + h(\mathbf{p},\mathbf{1}/n)
$$

where $r(\mathbf{w})$ was neglected since it does not involve $\mathbf{p}$. Note the expression of $h(\mathbf{p},\mathbf{1}/n) = \lambda\sum_i p_i\log(np_i) = \lambda\sum_i p_i\log(p_i) + \lambda\log(n)$ due to $\sum_i p_i = 1$. There are three constraints to handle, i.e., $p_i \geq 0, \forall i$ and $p_i \leq 1, \forall i$ and $\sum_i p_i = 1$. Note that the constraint $p_i \geq 0$ is enforced by the term $p_i\log(p_i)$, otherwise the above objective will become infinity. As a result, the constraint $p_i < 1$ is automatically satisfied due to $\sum_i p_i = 1$ and $p_i \geq 0$. Hence, we only need to explicitly tackle the constraint $\sum_i p_i = 1$. To this end, we define the following Lagrangian function

$$
L_{\mathbf{w}}(\mathbf{p},\mu) = -\sum_{i=1}^{n} p_i\ell(\mathbf{w};\mathbf{z}_i) + \lambda(\log n + \sum_i p_i\log(p_i)) + \mu(\sum_i p_i - 1)
$$

where $\mu$ is the Lagrangian multiplier for the constraint $\sum_i p_i = 1$. The optimal solutions satisfy the KKT conditions:

$$
- \ell(\mathbf{w};\mathbf{z}_i) + \lambda(\log(p_i^*(\mathbf{w})) + 1) + \mu = 0,
$$

$$
\sum_i p_i^*(\mathbf{w}) = 1
$$

From the first equation, we can derive $p_i^*(\mathbf{w}) \propto \exp(\ell(\mathbf{w};\mathbf{z}_i)/\lambda)$. Due to the second equation, we can conclude that $p_i^*(\mathbf{w}) = \frac{\exp(\ell(\mathbf{w};\mathbf{z}_i)/\lambda)}{\sum_i \exp(\ell(\mathbf{w};\mathbf{z}_i)/\lambda)}$. Plugging this optimal $\mathbf{p}^*(\mathbf{w})$ into the original min-max objective, we have

$$
\sum_{i=1}^{n} p_i^*(\mathbf{w})\ell(\mathbf{w};\mathbf{z}_i) - \lambda(\log n + \sum_i p_i^*(\mathbf{w})\log(p_i^*(\mathbf{w}))) + r(\mathbf{w}) = \lambda\log\frac{1}{n}\sum_i \exp(\ell(\mathbf{w};\mathbf{z}_i)/\lambda) + r(\mathbf{w}),
$$

which is the $F_{dro}(\mathbf{w})$ in the paper (the expression above Eq (2)).