# OpenReview forum: "An Online Method for A Class of Distributionally Robust Optimization with Non-convex Objectives"
_NeurIPS.cc/2021/Conference — NeurIPS 2021 Poster_

### Official Review · Reviewer_nTG1 · 2021-07-16

**Rating:** 5
**Confidence:** 5

**Summary:**

A new two sequence distributionally robust optimization algorithm and some modifications presented for a class of KL divergence regularized problems.


**Limitations And Societal Impact:**

Have addressed them appropriately.

**Main Review:**


Strength: + Online algorithm to solve compositional optimization problem in the context of deep learning.
+ Does not require to store the high dimensional variable p.

Weakness: - Writing can be improved quite a bit, and is some times unclear how exactly reformulations are defined or derived.


Justification: The overall idea is that the inner maximization problem, under some technical assumptions, can be solved in closed form. The paper mentions this but does not provide. This part of the paper is not fully clear -- -- the optimal solution of the inner maximization problem depends on w, and so it is not clear how we obtain F_dro from (1). In particular, it is not clear how r(w) changes from (1) to (2). I believe it is correct, essentially it follows from convex duality but there is no justification given in the paper.


It is very interesting to see that simple estimators such as the one proposed where they maintain two sequences to compute the gradient of the composite optimization problem can provide optimal convergence for the overall optimization problem. And moreover, the paper also shows how to exploit smooth in terms of PL condition to match the practical deep network instantiations of the training algorithm.

Experiments: The paper has fairly detailed experiments on a well studied classification task. The paper considers several other first order methods, and compares with some standard datasets including imbalanced datasets. The hyperparameters settings are described well and the algorithm may be implemented on standard deep learning frameworks by a practitioner from the description given in the paper. However, the experiments presented have nothing in connection with the theory presented. For example, the theoretical justifications presented are for the (expected) gradient norm and objective function values, whereas the experiments presented show convergence in test error. In this sense, in fact, (i) there are no experiments presented from an optimization perspective to valid the algorithm presented; (ii) even from the machine learning (generalization) perspective in the deep network context, the paper shows no experiments or even attempt to address some of the dependency relationship between the loss landscape and convergence, see https://papers.nips.cc/paper/2019/file/95e1533eb1b20a97777749fb94fdb944-Paper.pdf for example.





**Time Spent Reviewing:**

5

---

> ### Author Response · Authors · 2021-08-09
> **Thank you for your comments!**
>
> **Q1**: The overall idea is that the inner maximization problem, under some technical assumptions, can be solved in closed form. The paper mentions this but does not provide.
>
> **Response**: The derivation is not difficult to show. We provide a proof below. Thank you!
> Recall the problem: \begin{align*}
> \min_{\mathbf w\in\mathbb R^d}\max_{\mathbf p\in\Delta_n}F_{\mathbf p}(\mathbf w)= \sum_{i=1}^np_i\ell(\mathbf w; \mathbf z_i)  - h(\mathbf p, \mathbf 1/n) +  r(\mathbf w),
> \end{align*}
> where $\Delta_n =  \\{\mathbf p\in\mathbb R^n: \sum_i p_i=1, 0\leq p_i\leq 1\\}$.  In order to solve the inner maximization, we will fix $\mathbf w$ and derive an optimal solution $\mathbf p^*(\mathbf w)$ that depends on $\mathbf w$. To this end, we consider the following problem:
> \begin{align*}
> \min_{\mathbf p\in\Delta_n} -\sum_{i=1}^np_i\ell(\mathbf w; \mathbf z_i)  +  h(\mathbf p, \mathbf 1/n)
> \end{align*}
> where $r(\mathbf w)$ was neglected since it does not involve $\mathbf p$. Note the expression of $h(\mathbf p, \mathbf 1/n)=\lambda\sum_i p_i \log (np_i) = \lambda\sum_i p_i \log (p_i) + \lambda \log (n)$ due to $\sum_i p_i=1$. There are three constraints to handle, i.e., $p_i\geq 0, \forall i$ and $p_i\leq 1, \forall i$ and $\sum_i p_i=1$.  Note that the constraint $p_i\geq 0$ is enforced by the term $p_i\log(p_i)$, otherwise the above objective will become infinity. As a result, the constraint $p_i<1$ is automatically satisfied due to $\sum_ip_i=1$ and $p_i\geq0$. Hence, we only need to explicitly tackle the constraint $\sum_ip_i=1$. To this end, we define the following Lagrangian function
> \begin{align*}
>  L_{\mathbf w}(\mathbf p, \mu) = -\sum_{i=1}^np_i\ell(\mathbf w; \mathbf z_i)  +  \lambda(\log n + \sum_i p_i \log (p_i)) + \mu(\sum_i p_i - 1)
> \end{align*}
> where $\mu$ is the Lagrangian multiplier for the constraint $\sum_ip_i=1$. The optimal solutions satisfy the KKT conditions:
> \begin{align*}
>     &- \ell(\mathbf w; \mathbf z_i)  +  \lambda (\log (p^*_i(\mathbf w)) + 1) + \mu  = 0, \\
>     &\sum_ip^*_i(\mathbf w)=1
> \end{align*}
> From the first equation, we can derive $p^*_i(\mathbf w) \propto \exp(\ell(\mathbf w; \mathbf z_i)/\lambda)$, which together with
> the second equation yields $p^*_i(\mathbf w) = \exp(\ell(\mathbf w; \mathbf z_i)/\lambda)/\sum_i \exp(\ell(\mathbf w; \mathbf z_i)/\lambda)$.
>
> Plugging this optimal $\mathbf p^*(\mathbf w)$ into the original min-max objective, we have
>
> \begin{align*}
>      \sum_{i=1}^np^*_i(\mathbf w)\ell(\mathbf w; \mathbf z_i)  - \lambda(\log n + \sum_i p_i^*(\mathbf w) \log (p_i^*(\mathbf w)) ) +  r(\mathbf w) = \lambda \log \frac{1}{n}\sum_i \exp(\ell(\mathbf w; \mathbf z_i)/\lambda) + r(\mathbf w),
> \end{align*}
>
> which is the $F_{dro}(\mathbf w)$ in the paper (the expression above Eq (2)).
>
> **Q2**: However, the experiments presented have nothing in connection with the theory presented.
>
> **Response**: We partially disagree.  Please note that our experiments in part 1
> “Comparison with SOTA DRO Baselines” serve as the purpose to verify the convergence speed of the proposed algorithm in comparison with other optimization methods. The results shown in Figure 1 and Figure 2 clearly verify that our method converges faster and requires much less training time. Plotting the objective’s value or loss vs iterations or running time can be easily added without affecting our conclusion.

---

> > ### Comment · Reviewer_nTG1 · 2021-09-01
> > **Thanks for response**
> >
> > The derivation provided is correct, I appreciate providing the details.
> > Re experiments: It is true that the gradient norm is an optimization theoretic construction, and so it does not necessarily say a lot about nonconvex learning models considered here. However, the applications considered seem quite sensitive, and so I consider gradient norm to be a meaningful proxy, say for robustness of predictions, finetuning networks etc.. Moreover, it also gives an idea of what to expect when the  proposed method is used for applications or tasks outside of the experimental setups considered here.
> > Hence, I have raised my score.

---

> > > ### Author Response · Authors · 2021-09-01
> > > **Thank you for increasing your score!**
> > >
> > > Dear Reviewer nTG1:
> > >
> > > Thank you for increasing your score! We appreciate it.   We will include the convergence of gradient norm in the revision, which is a minor issue.
> > >
> > > Authors

---

### Official Review · Reviewer_Hg8V · 2021-07-17

**Rating:** 5
**Confidence:** 5

**Summary:**

This work proposes a practical online optimization method COVER and RECOVER for solving DRO with f-divergence, see eq(1).  They propose an online algorithm (RECOVER) for solving a more general formulation, by recasting the formulation into a composition minimization problem via fully exploiting the hidden structure. Moreover, they present the convergence result for the proposed method under the PL condition. Finally, the authors conduct experiments for several deep learning problems.

**Ethics Review Area:**

["I don’t know"]

**Main Review:**



In this paper, the authors consider an online method for distributionally robust optimization, which gets rid of the difficulty of high dimensional dual variable, which is usually used in the primal-dual type methods. The paper is overall well written and easy to follow. The main concerns raised by the reviewer are listed as follows:


1. The title misleads the contents of the paper. The paper shows a particular regularized version of distributionally robust optimization (DRO) with f-divergence--which is not the general DRO problem, and develops a online method for this. Although the studied problem (the author claimed) is a generalization of the DRO problem, it seems that the problem eq(1) in this paper is a relaxed/ regularized form, which may sacrifice the advantages of DRO’s modeling power. Actually, the algorithm scheme developed in this paper doesn't actually really tackle DRO in the constrained case (i.e. when one constrains $D_{KL}(P||P_0) \leq \delta$, rather than a regularization-based formulation), which is the hard case and the case that most researchers in DRO actually consider.

In details, I think the paper [11] cannot support your claim. It shows that the DRO formulation is equivalent to a variance-based regularization form in an asymptotic sense. Compared to the formulation (1) in Namkoong & Duchi, 2016, the considered formulation in eq (1) penalizes the divergence measure into the objective function and adds a convex regularizer. Then, are these two formulations equivalent? We can only obtain the equivalence result under certain conditions, see the reference [1, section 4] for details.

[1] Friedlander M P, Tseng P. Exact regularization of convex programs[J]. SIAM Journal on Optimization, 2008, 18(4): 1326-1350.

2. The author claims that the proposed method can be applied to eq(3) with a non-smooth regularizer, the theoretical analysis using the PL conditions requires the regularization term to be smooth. In Assumption 3, the gradient notion has been invoked in the PL inequality. Could you please elaborate the details how your analysis can be trivially extended to non-smooth problems?


**Time Spent Reviewing:**

3 hours

---

> ### Author Response · Authors · 2021-08-09
> **Thank you for your comments!  Please read our responses below for addressing the misunderstanding of our paper.**
>
> Thank you for your comments! We believe there might be some misunderstanding of our claims in the paper that are deemed to oversell our method by the reviewer. We request the reviewer to re-consider the difficulty for solving the considered KL regularized DRO formulation and our efforts in establishing the a state-of-the-art complexity and in extensive experiments  for evaluating our paper.  We also believe the proposed method could bring a different perspective for solving the constrained DRO problems.
>
> **Q1**: The title misleads the contents of the paper.
>
> **Response**: Please notice that our title is “An Online Method for **A Class of** Distributionally Robust Optimization with Non-convex Objectives”, where we explicitly mentioned for “a class of distributionally robust optimization”. In addition, in the abstract we have clearly remarked that we are considering a KL-regularized DRO formulation.
>
> **Q2**: Although the studied problem (the author claimed) is a generalization of the DRO problem, it seems that the problem eq(1) in this paper is a relaxed/ regularized form, which may sacrifice the advantages of DRO’s modeling power.
>
> **Response**: Please notice that we **did not claim** our studied problem is a generalization of the DRO problem from the modeling perspective. Instead, our words in section 3 are “For more generality, we consider the stochastic compositional problem …”, which mean that we consider a general form in Eq. (4), which is **more general than the considered DRO formulation** in Eq. (2). We also would like to point out that to the best of our knowledge there is no previous study that can solve the KL-regularized DRO formulation for large-scale deep learning problems efficiently (e.g., iNaturalist2019 data with 265,213 images, ImageNet-LT with 115.8K images). Our method on iNaturalist2019 data can **save days of training time** than existing method (please refer to Figure 1 right).
>
> **Q3**: Actually, the algorithm scheme developed in this paper doesn't actually really tackle DRO in the constrained case, which is the hard case and the case that most researchers in DRO actually consider.
>
> **Response**: We agree the constrained DRO formulation is harder, which is not our goal of this paper. We will add some discussion in the revision. However, designing a practical and scalable algorithm for a KL-regularized DRO formulation itself is not a trivial task. We would like to cite a recent NeurIPS 2020 paper by Levy, Carmon, Aaron Sidford (cf. reference [22] in the paper) to further demonstrate the significance of our result, which is also discussed in lines 100-101. They considered different formulations of DRO, which also includes our considered KL-regularized DRO formulation. Their assume that the loss function is convex and provide a sample complexity for the KL-regularized DRO formulation $O(1/\epsilon^3)$ for their method.  In contrast, we provide a better sample complexity in the order of $O(1/\epsilon)$ under a PL condition without convexity assumption. Additionally, their method requires a large batch size in the order of $O(1/\epsilon)$, while our method only requires a constant batch size which is more practical.
>
> **Q4**: Compared to the formulation (1) in Namkoong & Duchi, 2016, the considered formulation in eq (1) penalizes the divergence measure into the objective function and adds a convex regularizer. Then, are these two formulations equivalent?
>
> **Response**: We did not claim the constrained formulation in Namkoong & Duchi, 2016 is equivalent to the regularized formulation in our paper. Our original sentence is “It has been shown that for a family of divergence functions h(p, 1/n), different DRO formulations are statistically equivalent to a certain degree [11]”.   We cite [11] to support our claim that different divergence functions (in the constrained formulation) are statistically equivalent.
>
> **Q5**: The author claims that the proposed method can be applied to eq(3) with a non-smooth regularizer, the theoretical analysis using the PL conditions requires the regularization term to be smooth.
>
> **Response**:  In Section 5 when considering PL condition, we have explicitly said (lines 188-189): “In order to analyze RECOVER, we assume the following PL condition of the objective with **a differentiable regularization r** [44]”. We only consider non-smooth regularizer in section 4 for more generality of the proposed COVER algorithm. The update in step 6 in Algorithm 1 can handle non-smooth regularizer. In Section 5, when we consider PL condition, we assume a smooth regularizer (cf. lines 191 -192).

---

> > ### Author Response · Authors · 2021-09-01
> > **To Reviewer Hg8V: Please let us know if you have any further concerns and suggestions!**
> >
> > Dear Reviewer Hg8V,
> >
> > Thank you for your time for reading our comments. We hope that you have read our response.  Please let us know if you have any further concerns and suggestions. Although your current score tends to reject our paper for NeurIPS, we hope that you acknowledge the merits of our work towards addressing the computational challenge for solving DRO problems. We would like to summarize our problems and contributions for your information:
> > 1. **Importance of considered DRO formulation**: We consider a family of DRO problems with a KL regularizer on the dual variables. We admit that the constrained DRO formulation is more challenging (which is not our goal in this paper). However, there is still lack of efficient solutions for solving the considered DRO problems. It is also important to have an efficient solution for the considered regularized DRO formulation as (i) it  has more efficient solution (as provided by ours) than  the constrained DRO formulations; (ii) our experiments have demonstrated its effectiveness compared with empirical risk minimization for deep learning problems.
> >
> > 2. **Computational Challenge**: For the considered regularized DRO formulation, most previous algorithms that are based on primal dual algorithms suffer from a high per-iteration cost dependent on the data size. Recent works (e.g., Levy et al. NeurIPS 2020) have also considered the challenge for solving our considered DRO formulation. However, their solution is not as practical as ours since they require a large batch size and also suffer from a worse convergence rate.  In contrast, our result of achieving the optimal rate of $O(1/(\mu\epsilon))$ under a PL-condition is novel. In order to establish the improved rate, we have innovations in twofold (i) algorithmic level, we utilize the variance reduction techniques at the inner and outer level without using mega large mini-batch size at any iterations; (ii) proof level, we innovatively prove that the estimation error of the two sequences, u and v, are decreasing geometrically after a stage (Lemma 3). Please also notice the generality of our algorithms, which can be used for solving  a broad family of two-level compositional problems.
> >
> > 3. **Extensive Experiments**: We have conducted extensive experiments on benchmark and large-scale datasets in terms of both the comparison of convergence speed and generalization performance for deep learning with imbalanced data. The experiments have clearly demonstrated the effectiveness of our method.
> >
> > Overall, our paper has both significant  theoretical and empirical contributions, which we believe deserves a publication.  Given that there is increasing interest in addressing the computational challenge of DRO and in employing it for solving various deep learning problems, we believe our solution has brought some new perspectives to the community.
> >
> > We are kind of disappointed that you did not mention anything related to our algorithmic design and theoretical analysis.  We hope that you have noticed the merits of our work mentioned above. Your current score "4: Ok but not good enough - rejection" was based on your misunderstanding of our work that we plan to address constrained DRO problems. Other reviewers including pWxx,  WK9G,  and nTG1 all appreciate the algorithmic design and the practical method that we delivered. Your non-response to our rebuttal seems to indicate that you do not have any concerns of our work. But your reluctance to respond to our rebuttal in the rolling discussion phase and high confidence score would give the AC the wrong impression and would also make it difficult for the AC to make the decision since we also received a  high rating score of 7 from reviewer pWxx. Hence, we request you to check our response again, and acknowledge the merits of this paper or point out any other concerns that we could address in the revision. This is very important to maintain a healthy reviewing system that would benefit the community and advance the field  as we believe that you are as serious as us for advancing the field.
> >
> > Thank you!
> >
> > Authors

---

> > > ### Comment · Reviewer_Hg8V · 2021-09-02
> > > **Response**
> > >
> > > Thanks for your patience and comprehensive response. As pointed out by other reviewers and authors,  the constrained DRO problem is still rather computationally demanding.  The authors provide an interesting solution to this problem. I would be rather happy to raise my score to 5. However, there exist some points that can be further improved.
> > > 1. The regularized parameter $\lambda$ --- I would suggest the authors add its sensitivity experiment. As pointed out by my early review, I think it is hard to find the equivalent correspondence. Instead, it is better to justify the modeling power empirically.  I read these sentences as a manner to claim some novelty with that respect which I think is incorrect.
> > > 2. “We only consider non-smooth regularizer in section 4 for more generality of the proposed COVER algorithm.” To avoid the misunderstanding, it is better to explicitly caution the reader in section 4, e.g., change the Assumption 3(c) or add a remark. As your algorithm used proximal mapping, it is natural to make the reader misunderstand there is a non-smooth term to address. However, the analysis provided in the work cannot handle the case.

---

> > > > ### Author Response · Authors · 2021-09-02
> > > > **Thank you for your comments! You missed our related results. Please read our response and check our paper.**
> > > >
> > > > Dear Reviewer Hg8V,
> > > >
> > > > Thank you for raising your score. However, your might have missed our results in the paper related to the two points that you mentioned:
> > > >
> > > > 1. **Sensitivity experiment is included in the paper**. An experiment about sensitivity of $\lambda$ is included in Figure 3. We included this result according to a reviewer's suggestion for our earlier submission to ICML. The modeling capabilities of the proposed method have been demonstrated in Table 2 and Table 3 comparing with standard deep learning methods.
> > > >
> > > >
> > > > 2. **Proximal Gradient Convergence is in Theorem 2**. Thank you for your suggestion. We will make it more clear in the paper. We would still like to point out that our analysis in Section 3 is more general covering the non-smooth regularizer, where Theorem 2 establishes  the convergence for first-order convergence in terms of proximal gradient.
> > > >
> > > > We believe that the above two issues mentioned above are either already addressed or an minor issue.
> > > >
> > > > Thank you for your consideration!
> > > >
> > > > Authors

---

### Official Review · Reviewer_pWxx · 2021-07-19

**Rating:** 7
**Confidence:** 2

**Summary:**

This paper studies the family of distributionally robust optimization (DRO) problems with non-convex objectives and KL-divergence. With the motivation of improving the computational efficiency on existing methods and applying to online settings, the authors proposed a new duality-free online stochastic methods for solving the considered DRO problem. Theoretical analyses on complexities are provided for the proposed algorithm with and without a PL condition. Comparisons with state-of-the-art DRO methods are presented over ERM on imbalanced datasets.

**Limitations And Societal Impact:**

Yes

**Main Review:**

The paper has a clear motivation of improving the complexities of existing DRO methods from a theoretical perspective. Comparisons are summarized clearly in Table 1 in the related work section. The structure of the paper is generally easy to follow, though many parameters and variables are defined to simplify the arguments. I appreciate that the authors lay out the list of required assumptions in Section 3. Overall, I think this paper is written with good quality and has theoretical contributions to the field of DRO.

One question I have for the paper is regarding the algorithmic design of COVER in Section 4. I have a hard time understand why the update rules of u and v designed in line 7 of Algorithm 1 lead to a better convergence result, compared with existing state-of-the-art methods for DRO. Can you briefly discuss the motivations for this algorithmic design?

Moreover, I think the notations introduced in this paper seem to be a little bit complicated. It would be better to include a table to summarize the core parameters. Other suggestions regarding the paper are listed below:

1.	It would be better to introduce some specific examples of typical DRO problems, apart from the general framework. This will help readers better understand the assumptions and the user cases for the proposed framework.

2.	As for the experiments, is there any reason why you only focus on the imbalanced datasets? It would be great if you can provide empirical studies on other types of DRO problems, such as noisy data and adversarial data.


**Time Spent Reviewing:**

3 hours

---

> ### Author Response · Authors · 2021-08-09
> **Thank you for the positive rating and suggestions!**
>
> Q1: One question I have for the paper is regarding the algorithmic design of COVER in Section 4. I have a hard time understand why the update rules of u and v designed in line 7 of Algorithm 1 lead to a better convergence result, compared with existing state-of-the-art methods for DRO. Can you briefly discuss the motivations for this algorithmic design?
>
> **Response**: The update rule of u and v designed in line 7 has a variance-reduction property, which is a key factor that enables us to achieve faster convergence.  Eq (7) in Lemma 3 can help demonstrate the variance-reduction property. These two updates are motivated from existing variance-reduction techniques for estimating the stochastic gradient [8, 12, 13]. For our problem, the key is to estimate the gradient of the compositional function, i.e., $\nabla g(w)^{\top}\nabla f(g(w))$. Since $f$ is a deterministic function and $g$ is a stochastic function, we need to estimate $\nabla g(w)$ and $g(w)$ by a mini-batch of samples. Hence, we leverage the variance reduction updates to track both $\nabla g(w)$ and $g(w)$ by $v$ and $u$ to enjoy faster convergence. Some existing method, e.g., [5] only estimates $g(w)$ by the same variance reduction technique and estimates $\nabla g(w)$ simply by a  mini-batch gradient. Hence, their algorithm has a worse complexity in the order of $O(1/\epsilon^2)$ for finding a solution satisfying $\mathbb E\|\nabla F(w)\|^2\leq \epsilon$, which is worse than our complexity of $O(1/\epsilon^{3/2})$ (in section 4.1). Compared with existing state-of-the-art methods for DRO, there are several perspectives to be considered: (i) per-iteration costs; (ii) practicability; and (iii) convergence rate. For this comparison, we refer the reviewer to Table 1.
>
> Q2: It would be better to introduce some specific examples of typical DRO problems, apart from the general framework. This will help readers better understand the assumptions and the user cases for the proposed framework.
>
> **Response** : Thank you for the suggestions. One use case of DRO is what we considered in the experiments about learning with imbalanced data. Another use case of our method is for learning with noisy data, e.g., [4].  We will provide more details in the revision.
>
> Q3: As for the experiments, is there any reason why you only focus on the imbalanced datasets?
>
> **Response**: There is no particular reason other than that it is an interesting and importance use case of the proposed method.

---

### Official Review · Reviewer_WK9G · 2021-07-21

**Rating:** 6
**Confidence:** 2

**Summary:**

This paper propose an online algorithm to solve a class of Distributionally Robust Optimization problem with non-convex objective functions. The standard primal-dual method typically suffer from optimization over high dimensional variable, which is impractical in many real world applications. The sota complexity of proposed method is provided with and without the PL condition. Empirical experiments demonstrate the superiority of the proposed algorithm.

**Limitations And Societal Impact:**

Yes.

**Main Review:**

This paper is in general well written, the complexity analyses and numerical experiments are seems to be promising. I only have some comments related to the model.

1. The problem studied in this paper seems to be slightly different from the standard literature. For example, the problem in [1] aims to maximize probability distribution p over a KL-ball center at the empirical distribution, while the problem in this paper maximize p over probability simplex. As a result, the \lambda in eq(2) is a fixed parameter, while in [1] we still need to optimize over \lambda. So, intuitively, the problem in this paper should be easier to solve compare to standard KL-based DRO problem. From the modeling prospective, the model in this paper is less informative if we want to understand the relationship between regularization parameter \lambda and the level of distributional robustness.

2. Have the authors thought about generalizing the current framework to Wasserstein-based DRO (where the inner maximization typically do not have a closed form representation)? What would be the main challenges for doing so?

**Time Spent Reviewing:**

2

---

> ### Author Response · Authors · 2021-08-09
> **Thank you for your comments!**
>
> **Q1**: So, intuitively, the problem in this paper should be easier to solve compare to standard KL-based DRO problem.
>
> **Response**: We agree that our considered problem is relatively easier than the DRO formulation with a KL ball constraint. However, even for our considered formulation there is **no efficient and practical** algorithm that can be scalable to large-scale data for deep learning. In particular, most existing studies focus on primal-dual methods, which has a per-iteration cost of $O(n)$ with $n$ being the size of training set. In experiments, we have observed that on the large-scale iNaturalist dataset with 265,213 images. Our method is significantly faster than the primal-dual methods (stoc-AGDA [43], PG-SMD2 [35] and PES-SGDA [14]) by saving several days of training time (please refer to Figure 1 right).
>
> We would like to cite a recent NeurIPS 2020 paper by Levy, Carmon, Aaron Sidford (cf. reference [22] in the paper) to further demonstrate the significance of our result, which is also discussed in lines 100-101. They considered stochastic optimization for several formulations of DRO, which also includes our considered KL-regularized DRO formulation. Their assume that the loss function is convex and provide a sample complexity for solving the KL-regularized DRO formulation in the order of $O(1/\epsilon^3)$ of their method.  In contrast, we provide a better sample complexity in the order of $O(1/\epsilon)$ under a PL condition without the convexity assumption. Additionally, their method requires a large batch size in the order of $O(1/\epsilon)$ that is not practical, while our method only requires a constant batch size.
> We request the reviewer to take the difficulty of optimizing the considered KL-regularized DRO formulation into account to re-evaluate the significance of our results. Besides, to the best of our knowledge, this is the first work that makes DRO practical for deep learning as demonstrated in the paper.
>
> **Q2**: From the modeling prospective, the model in this paper is less informative if we want to understand the relationship between regularization parameter \lambda and the level of distributional robustness.
>
> **Response**: Agree. But we focus on addressing the optimization challenge in this paper. In addition, in practice no matter whether you use regularized form or constrained form, one has to tune the involved parameters. Our experiments in Section 6 for deep learning (cf Table 2 and Table 3, including a result for a large-scale ImageNet-LT data with 115.8K images) have demonstrated the effectiveness of the proposed method. Without a scalable and practical algorithm for solving DRO, such a development is difficult.
>
> **Q3**: Have the authors thought about generalizing the current framework to Wasserstein-based DRO (where the inner maximization typically do not have a closed form representation)? What would be the main challenges for doing so?
>
> **Response**: While we can formulate a min-max problem into a compositional formulation, the main challenge is that the outer function of the compositional formulation might be difficult/impossible to evaluate depending on what Wasserstein distance is used.

---

### Decision · Program_Chairs · 2021-09-27

**Decision:**

Accept (Poster)

**Comment:**

Although there is some concern about how to best compare this work with prior efforts on robust optimization, overall it seems there are some interesting ideas and a new contribution presented that provides a more efficient alternative to standard distributionally robust methods.